# Bilateral human laryngeal motor cortex in perceptual decision of lexical tone and voicing of consonant

Baishen Liang [1,2], Yanchang Li [1], Wanying Zhao[1] & Yi Du [1,2,3,4] ✉

Speech perception is believed to recruit the left motor cortex. However, the exact role of the laryngeal subregion and its right counterpart in speech perception, as well as their temporal patterns of involvement remain unclear. To address these questions, we conducted a hypothesis-driven study, utilizing transcranial magnetic stimulation on the left or right dorsal laryngeal motor cortex (dLMC) when participants performed perceptual decision on Mandarin lexical tone or consonant (voicing contrast) presented with or without noise. We used psychometric function and hierarchical drift-diffusion model to disentangle perceptual sensitivity and dynamic decision-making parameters. Results showed that bilateral dLMCs were engaged with effector specificity, and this engagement was left-lateralized with right upregulation in noise. Furthermore, the dLMC contributed to various decision stages depending on the hemisphere and task difficulty. These findings substantially advance our understanding of the hemispherical lateralization and temporal dynamics of bilateral dLMC in sensorimotor integration during speech perceptual decision-making.

Speech perception has long been hypothesized to recruit motoric simulation by the speech motor system, as posited by the motor theory of speech perception[1]. Recent neuroanatomical models of speech processing propose that the left motor cortex maps phonological analyses onto motor representations, which may compensate for degraded auditory processing in challenging listening conditions[2–4]. Transcranial magnetic stimulation (TMS) studies have identified a causal engagement of the left motor cortex in speech perception in an effector-specific manner, such as the lip motor subregion for bilabial consonants[5,6], and the tongue motor area for dental consonants[6,7] and vowels[8], while the right motor cortex has been linked to non-lexical prosodic cues[9]. However, three outstanding questions regarding the role of the bilateral motor cortices in speech perception remain unanswered: (1) whether the laryngeal motor cortex (LMC) is engaged in an effector-specific manner similar to the lip and tongue areas, (2) how bilateral motor cortices cooperate during speech perception

under varying difficulty, and (3) what specific stages of the perceptual decision-making process the bilateral motor cortices modulate.

This study aims to address the first and second questions (spatial questions) by exploring four hypothetical mechanisms that may drive the functional distributions of bilateral motor cortices in speech perception: the acoustic hypothesis, the lexical hypothesis, the motor hypothesis, and the redundancy hypothesis (Fig. 1b–e). To do so, we delivered repetitive TMS (rTMS, Experiment 1) or theta-burst stimulation (TBS, Experiment 2, including intermittent TBS, i.e., iTBS, and continuous TBS, i.e., cTBS) to the left or right motor cortex (in Experiment 1, LMC and tongue motor cortex, TMC; in Experiment 2, the LMC only) of Mandarin speakers to investigate if the identification of lexical tone (Tone1 vs. Tone2, featured by pitch contour) and dental plosive consonant ([t] vs. [tʰ], featured by voice onset time, VOT) in quiet or in noisy background would be modulated accordingly (Fig. 1f–h). To localize the dorsal LMC (dLMC), which is closely related

[1]Institute of Psychology, CAS Key Laboratory of Behavioral Science, Chinese Academy of Sciences, Beijing 100101, China. [2]Department of Psychology, University of Chinese Academy of Sciences, Beijing 100049, China. [3]CAS Center for Excellence in Brain Science and Intelligence Technology, Shanghai 200031, China. [4]Chinese Institute for Brain Research, Beijing 102206, China. ✉ e-mail: duyi@psych.ac.cn

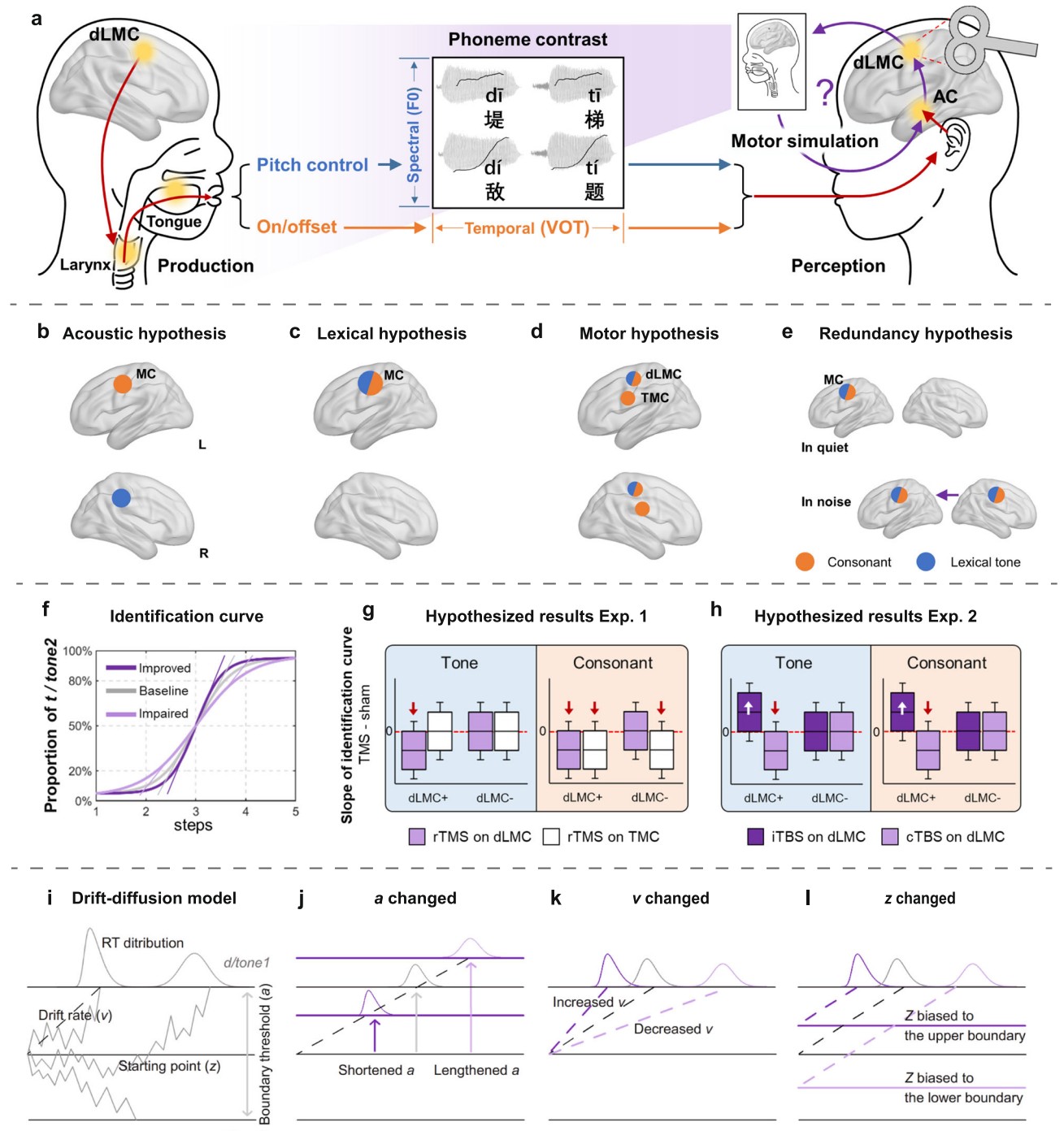

to speech production[10], and the TMC, respectively, participants underwent a functional magnetic resonance (fMRI) pretest where they performed phonation and tongue movement tasks.

The acoustic hypothesis posits that bilateral motor cortices process speech in ways resembling the auditory cortices, with the left motor cortex being more sensitive to temporal modulation and the right counterpart more attuned to spectral modulation[11,12]. If so, the perception of consonant with temporally fast-varying VOT is expected to be left-lateralized, whereas that of lexical tone with spectrally fast-changing contour is expected to be right-dominant (Fig. 1b). In contrast, the lexical hypothesis suggests that the left motor cortex is more involved in lexical processing, and hence left lateralization is expected for both lexical tone and consonant perception (Fig. 1c). The motor hypothesis suggests that the motor cortices generate an "internal

model" in speech perception as if they are enrolled in articulation with bilateral effector specificity[13,14]. Since lexical tones are determined by laryngeal movements (pitch regulations) and articulating dental plosives recruit both the larynx (voicing on/offset) and tongue (dental consonants)[15], based on the motor hypothesis, lexical tone perception would engage the dLMC, whereas consonant perception would enroll both the dLMC and TMC (Fig. 1d). Finally, the redundancy hypothesis proposes that the left motor cortex is more fundamental in speech perception, with its right counterpart being redundant and optimized only when the left is insufficient for completing perception (Fig. 1e). Lesions[16] or "virtual lesions"[17,18] studies have shown that perturbation of the left language areas triggers compensatory activations in the right counterparts. Background noise was added to half of the conditions and acoustic continuum was used to estimate the redundancy

**Fig. 1 | Experimental design and hypothesized results. a** Schematic illustration of sensorimotor integration in Mandarin speech perception. The dLMC commands the larynx to control the pitch of voice and onset/offset of phonation during speech articulation. The dLMC is assumed to engage in the perception of lexical tone and consonant (VOT contrast) in Mandarin listeners, in the way of motor simulation as in speech production. **b–e** Predicted spatial patterns of bilateral motor engagement by four different hypotheses: the acoustic hypothesis (**b**), the lexical hypothesis (**c**), the motor hypothesis (**d**), and the redundancy hypothesis (**e**). See the main text for explanations of each hypothesis. **f–h** Hypothetical results for psychometric curve fitting. **f** How TMS may affect curves of identification: compared with baseline (gray line), inhibitory stimulation (rTMS and cTBS) would "flatten" the curve and impair performance (light purple line), while excitatory stimulation (iTBS) would "steepen" the curve and improve performance (dark purple line). Oblique lines crossing the point of subjective equality (PSE) represent the slopes of the curves. We used the slope estimation method to test the hemispheric asymmetry and effector-specificity of motor engagement. **g, h** How slopes of tone and consonant curves may be altered by TMS in Experiment 1 (**g**) and

Experiment 2 (**h**) if (+) or if not (−) the dLMC is engaged, compared with sham (red dashed lines). Red arrows: inhibitory effect; white arrows: excitatory effect. The boxplots in (**g**) and (**h**) are schematic representations of possible slope distributions for ease of reading, while the centers, bounds of the box, and whiskers only represent the data distribution for the simulation. **i–l** Temporal stages of motor engagement were tested by the hierarchical drift-diffusion model (HDDM). **i** How HDDM predicts distributions of reaction times. **j–l** Hypothetical models with the parameter $a$ (boundary threshold, **j**), $v$ (drift rate, **k**), and $z$ (starting point, **l**) modulated by TMS, respectively. For the whole figure: blue and orange represent lexical tone and consonant, respectively; light purple, dark purple, white, and gray represent inhibitory TMS (rTMS and cTBS), excitatory TMS (iTBS), TMC stimulation, and sham, respectively. These color patterns are the same in Figs. 2, 3. BrainNet Viewer was used to generate the schematic brain maps for (**a–e**). COPYRIGHT NOTICE: © Copyright 2007, NITRC. All rights reserved. [https://www.nitrc.org/include/copyright.php] MC motor cortex, AC auditory cortex, dLMC dorsal laryngeal motor cortex, TMC tongue motor cortex.

hypothesis by comparing the engagement of bilateral motor cortices under varying task difficulties.

For the third (temporal) question, speech perceptual decision is postulated as a three-stage procedure encompassing extraction of acoustic-phonetic features, mapping to phonemic categories, and response selection, in which the (left) motor cortex is interactively involved in all stages along with the auditory cortices[19]. Although less is known about the right motor cortex, we hypothesized that it may be involved in all temporal stages in speech perceptual decision due to the functional symmetry to the left counterpart and its subregions during articulation[13,14]. To test this, we applied the hierarchical Bayesian estimation of the drift-diffusion model (HDDM)[20,21] to single-trial binary responses and reaction times (RTs) in Experiment 2 to disentangle what latent dynamic decision processes the dLMC is engaged in (Fig. 1i–l).

Our results reveal an effector-specific involvement of bilateral dLMCs in the perceptual decision of both lexical tone and voicing of consonants, lending support to the motor hypothesis. Meanwhile, we provide evidence for the redundancy hypothesis, as the left dLMC plays a dominant role, while the right counterpart is only crucial in challenging tasks. In contrast, the lexical hypothesis is only weakly supported, whereas the acoustic hypothesis is not confirmed. Moreover, the specific perceptual decision stages that are modulated by the dLMC hinge on the hemisphere and task difficulty. Taken together, these findings expand our knowledge of the underlying mechanisms and temporal dynamics of bilateral motor engagement in speech perceptual decisions.

## Results

### The dLMC is involved in the perception of lexical tone and voicing of consonant

In Experiment 1, which is exploratory, we used rTMS to estimate changes in Mandarin lexical tone and consonant perception in a group of 64 young adults. Participants were divided into two matched groups for stimulation on either the left or right motor cortex. Experiment 1 included 3 rTMS sessions: sham, dLMC stimulation, and TMC stimulation. Within each session, participants underwent four blocks of syllable identification tasks that included tone and consonant tasks with or without noise masking, and were required to perform forced-choice identification judgments. Syllables were randomly selected from a 5×5-step tone–consonant continuum matrix (Fig. 2a). Starting with the syllable onsets, three pulses of 10-Hz biphasic rTMS were applied to the dLMC or the TMC (Fig. 2b). The TMC stimulation was set as a site control in answering whether the motor engagement is effector-specific (the first spatial question), and verifying the paradigm as the engagement of the TMC in dental consonant perception has been well recognized[6,7]. The MNI

coordinates of the dLMC [±40, −5, 50] and the TMC [±59, −3, 36] were defined by an fMRI localization pretest in 48 Mandarin speakers (see Supplementary Methods, Functional localization experiment). Both targets were located in the premotor cortex (Brodmann area 6), which is suggested to be a transfer node in the sensorimotor transforming pathways[23] and may subserve auditory-motor mapping in adverse listening conditions[24]. This localization approach optimized the effectiveness of TMS on our chosen tasks. The Euclidean distance between the two targets enabled the spatial dissociation of TMS effects on the dLMC and TMC[25]. Meanwhile, to avoid confounding the examination of the motor somatotopy, labial consonants were intentionally excluded, as they may activate the lip motor area, which is in close proximity to our targeted regions[13]. Note that, we only found activations in the dLMC but not the ventral LMC (Supplementary Fig. 4). From an evolutionary perspective, the human-specific dLMC has been found to control vocal pitch during speech and singing[10].

Slopes of psychometric functions were used to quantify the perceptual sensitivity for phoneme categorization[26] and the modulation effects by rTMS. Unexpectedly, we found competitions between tone and consonant perception such that slopes were affected by the ambiguity of the unattended dimension (see Supplementary Methods, Competitions between consonant and tone perception). To rule out this interference, for each block, slopes were separately extracted for trials with syllables being unambiguous, half-ambiguous, and ambiguous in the orthogonal dimension (see Methods, Experiment 1, Stimulation effect analyses).

Results showed that rTMS on the dLMC exerted inhibitory effects on both tone and consonant perception. Tone perception in noise was impaired by stimulating the left dLMC when the unattended consonant dimension was ambiguous (compared with sham: $p_{fdr} < 0.001$, Cohen's $d = -1.166$, permutation test; compared with TMC stimulation, $Z = -4.027$, $p_{fdr} < 0.001$, Cohen's $d = -1.086$, Wilcoxon signed rank test, Fig. 2m), and by stimulating the right dLMC when the unattended consonants were half-ambiguous (compared with TMC stimulation: $Z = -3.175$, $p_{fdr} = 0.009$, Cohen's $d = -0.838$, Wilcoxon signed rank test, Fig. 2j). Left dLMC stimulation also impaired consonant perception without noise masking when tones were unambiguous (compared with sham: dLMC, $p_{fdr} = 0.033$, Cohen's $d = -0.459$, permutation test, Fig. 2c). Meanwhile, consistent with previous studies[6,7,22], perception of dental consonant was impaired by stimulating the left TMC when tones were unambiguous and in quiet (compared with sham: $p_{fdr} = 0.033$, Cohen's $d = -0.467$, permutation test, Fig. 2c), and by stimulating the right TMC when tones were ambiguous and in noise (compared with sham: $p_{fdr} = 0.040$, Cohen's $d = -0.940$, permutation test, Fig. 2n), confirming the effectiveness of our paradigm. In contrast, rTMS on the TMC did not influence tone perception in any condition ($ps > 0.05$).

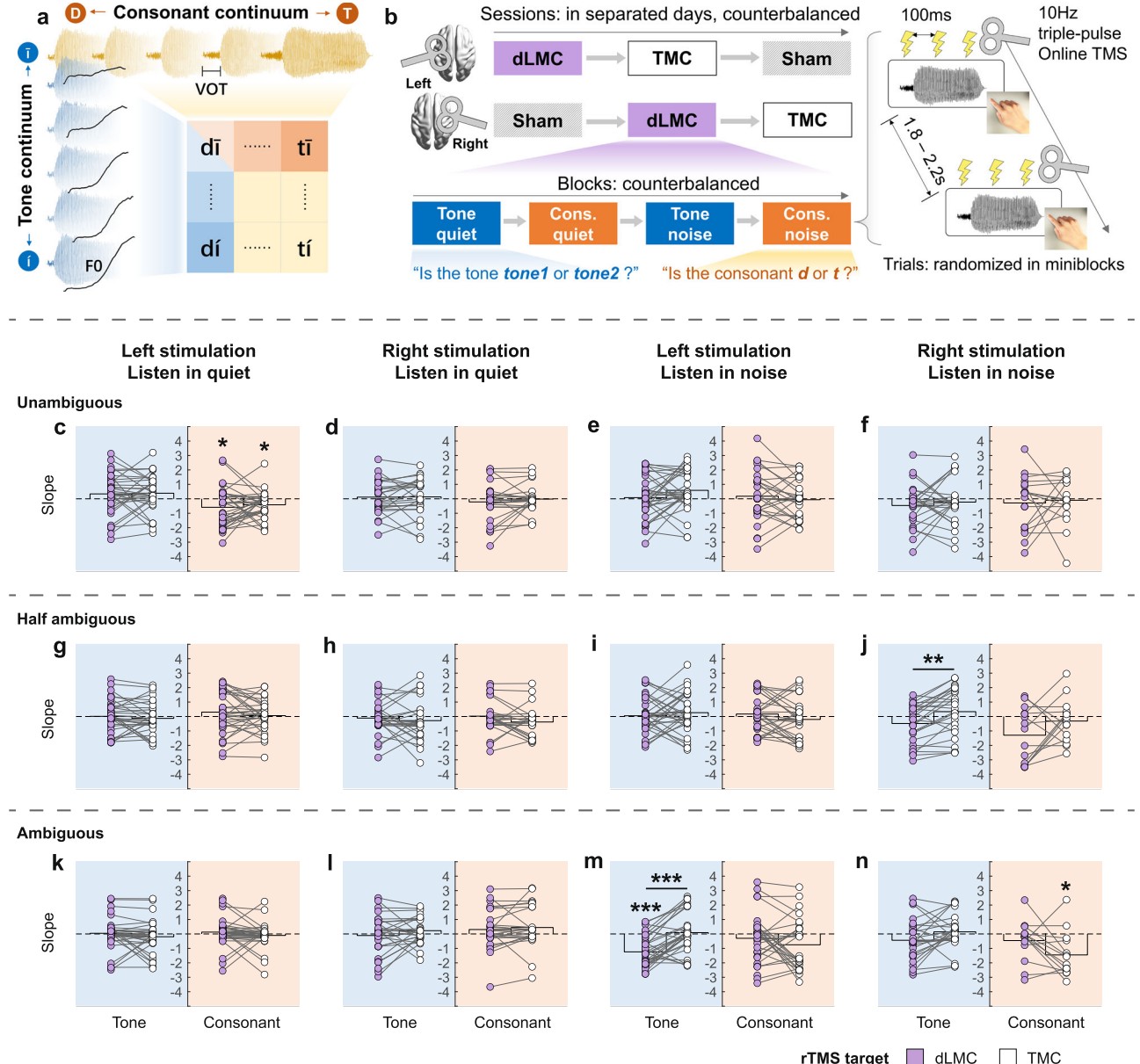

**Fig. 2 | Protocol and rTMS effects on slopes of identification curves in Experiment 1. a, b** Design of Experiment 1. **a** The orthogonal design of the tone–consonant stimuli matrix. **b** The experimental procedure at session, block, and trial levels. 10-Hz triple-pulse online rTMS was applied at the syllable onset in each trial. **c–f** When syllables were unambiguous in the unattended dimension, rTMS upon the left dLMC (compared with sham: $p_{fdr}$ = 0.033, **c**) and TMC impaired consonant perception in quiet (compared with sham: $p_{fdr}$ = 0.033, **c**). **g–j** When syllables were half-ambiguous in the unattended dimension, compared with right TMC stimulation, rTMS upon the right dLMC impaired tone perception in noise (compared with TMC stimulation: $p_{fdr}$ = 0.009, **j**). **k–n** When syllables were ambiguous in the unattended dimension, rTMS upon the left dLMC impaired tone perception in noise (compared with sham: $p_{fdr}$ < 0.001; compared with TMC stimulation, $p_{fdr}$ < 0.001, **m**) while stimulating the right TMC impaired consonant perception in noise (compared with sham: $p_{fdr}$ = 0.040, **n**). For each panel, dots represent log slopes of psychometric curves for each participant; bars show the group average of slopes in each task. Light purple dots represent slopes in

conditions where the dLMC was stimulated, whereas white dots are for slopes in TMC stimulation conditions. The blue background represents tone tasks, whereas the orange background represents consonant tasks. BrainNet Viewer was used to generate the schematic brain maps for (**b**). COPYRIGHT NOTICE: © Copyright 2007, NITRC. All rights reserved. [https://www.nitrc.org/include/copyright.php] dLMC: dorsal laryngeal motor cortex; TMC: tongue motor cortex. Sample sizes were equal across tasks and ambiguity conditions (left dLMC stimulation: 35; right dLMC: 28; left TMC: 33; right TMC: 26, individual participants), but the slopes were eliminated if the corresponding sham slopes were invalid (see the source data for Fig.2, Supplementary Table 2, and Supplementary Methods, Preprocessing of the slope). Statistical tests were performed by non-parametric tests comparing rTMS effects with zero (permutation test, null hypothesis TMS - Sham ≥0), and comparing rTMS effects upon dLMC with TMC within the same tasks (Wilcoxon signed rank test, null hypothesis dLMC - Sham ≥ TMC -Sham). P values (one-tailed) were adjusted by false discovery rate (FDR) correction (threshold = 0.05). *$p_{fdr}$ < 0.05; **$p_{fdr}$ < 0.01; ***$p_{fdr}$ < 0.001. Source data are provided as a Source Data file.

This validated the focality of rTMS and that tone perception is effector-specific to the dLMC.

Although Experiment 1 suggests the dLMC engagement in lexical tone and consonant perception, results are inconsistent and unreliable, as separating trials into three groups of ambiguity left a

limited number of trials for curve-fittings[26]. To verify the dLMC stimulation effects in Experiment 1, we applied offline TBS in another group of 26 participants in Experiment 2. TBS is a TMS paradigm whose directionality has been extensively explored, as iTBS and cTBS stimulation would increase and decrease cortical excitability,

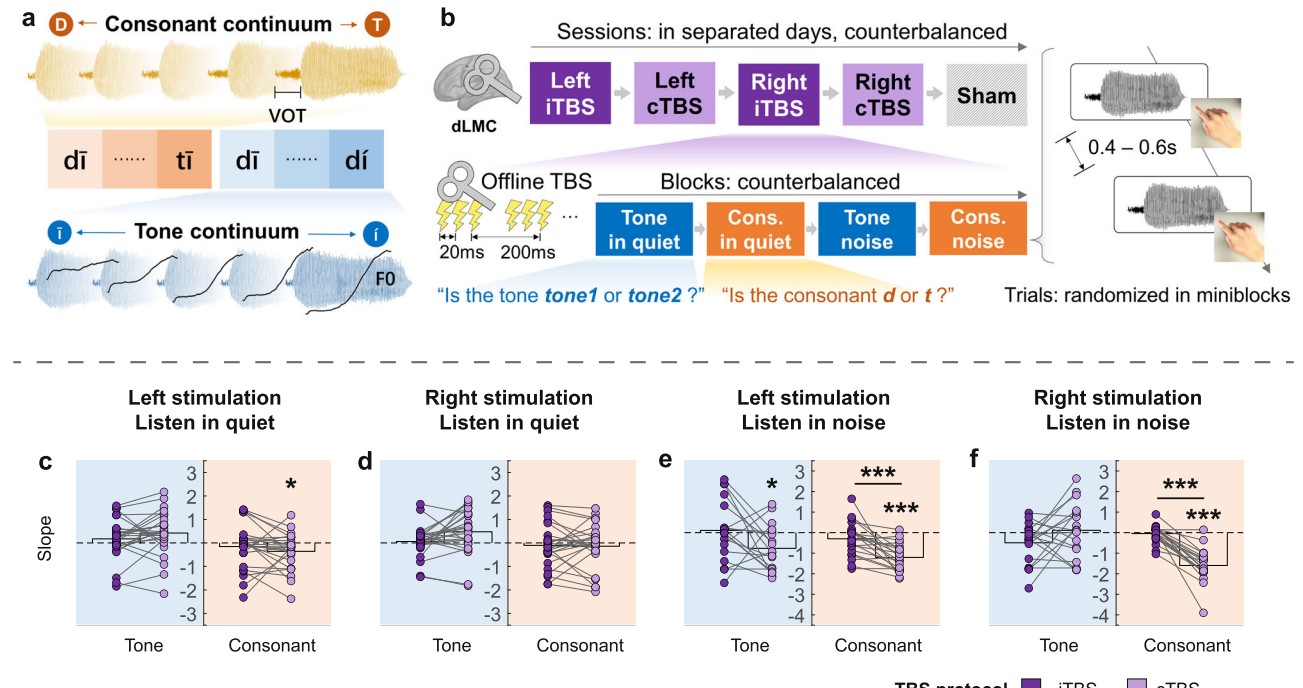

**Fig. 3 | Protocol and TBS effects on slopes of identification curves in Experiment 2. a, b** Design of Experiment 2. **a** The separate lexical tone and consonant continua. **b** The experimental procedure at session, block, and trial levels. Instructions were identical to Experiment 1. Offline iTBS or cTBS was applied to the left or the right dLMC before the task. **c** Compared with sham, cTBS upon the left dLMC impaired consonant perception in quiet ($p_{fdr}$ = 0.049). **d** TBS upon the right dLMC had no effect on tone or consonant perception in quiet ($ps > 0.05$). **e** cTBS upon the left dLMC impaired both tone (compared with sham, $p_{fdr}$ = 0.029) and consonant perception in noise (compared with sham, $p_{fdr} < 0.001$, and with iTBS, $p_{fdr} < 0.001$). **f** cTBS upon the right dLMC impaired consonant perception in noise (compared with sham, $p_{fdr} < 0.001$, and with iTBS, $p_{fdr} < 0.001$). See Fig. 2 legend for a detailed description of color patterns. BrainNet Viewer was used to generate the

schematic brain maps for (**b**). COPYRIGHT NOTICE: © Copyright 2007, NITRC. All rights reserved. [https://www.nitrc.org/include/copyright.php] Sample sizes were equal across tasks and stimulation conditions ($n$ = 25 participants), but the slopes were eliminated if the corresponding sham slopes were invalid (see the source data for Fig.3, Supplementary Table 2, and Supplementary Methods, Preprocessing of the slope). Statistical tests were performed by non-parametric tests comparing TBS effects with zero (permutation test, null hypothesis cTMS - Sham ≥ 0 and iTMS - Sham ≤ 0), and comparing effects of cTBS with iTBS within the same tasks (Wilcoxon signed rank test, cTBS – Sham ≥ iTBS – Sham). $P$ values (one-tailed) were adjusted by false discovery rate (FDR) correction (threshold = 0.05). *$p_{fdr} < 0.05$; **$p_{fdr} < 0.01$; ***$p_{fdr} < 0.001$. Source data are provided as a Source Data file.

respectively[27]. Moreover, to avoid competition between tone and consonant, in Experiment 2, we used one F0 and one VOT continuum, instead of a matrix, for the tone and consonant task, respectively (Fig. 3a). Furthermore, to directly estimate the hemispheric asymmetry of the dLMC in speech perception (the second spatial question), each participant received TBS on both hemispheres in five separated sessions in a random order: sham, iTBS on the left dLMC, cTBS on the left dLMC, iTBS on the right dLMC, and cTBS on the right dLMC (Fig. 3b). In each session, they performed four tasks resembling Experiment 1 after receiving TBS (Fig. 3b). Slopes of curves were analyzed as in Experiment 1. We hypothesized that iTBS and cTBS on the dLMC would improve and impair perception, respectively (Fig. 1h).

Results showed that cTBS on the dLMC inhibited both tone and consonant perception. Tone perception in noise was impaired by cTBS on the left dLMC (compared with sham: $p_{fdr}$ = 0.029, Cohen's $d$ = −0.688, permutation test, Fig. 3e). Consonant perception was impaired by cTBS on the left dLMC both when noise masking was absent (compared with sham: $p_{fdr}$ = 0.049, Cohen's $d$ = −0.474, permutation test, Fig. 3c) and present (compared with sham: $p_{fdr} < 0.001$, Cohen's $d$ = −1.918, permutation test; compared with iTBS, $Z$ = −3.715, $p_{fdr} < 0.001$, Cohen's $d$ = −1.030, Wilcoxon signed rank test, Fig. 3e). cTBS on the right dLMC also impaired consonant perception in noise (compared with sham: $p_{fdr} < 0.001$, Cohen's $d$ = −2.323; in comparison with iTBS, $Z$ = −3.901, $p_{fdr} < 0.001$, Cohen's $d$ = −1.970, Wilcoxon signed rank test, Fig. 3f) but did not affect consonant perception in quiet or tone

perception ($ps > 0.05$). However, iTBS did not exert any effect in any condition ($ps > 0.05$). Note that cTBS had a more extensive impact on consonant perception than on lexical tone perception. Despite our efforts to balance the individualized SNR levels for both tasks (see Methods, Stimuli presentation), plosive consonants (featured by the length of the aspiration noise resembling the masking noise) are less resilient to noise and more challenging to identify in noise (most participants performed unsatisfactorily in the consonant-in-noise perception task; see Supplementary Fig. 2). We also tested the cTBS effects on slopes after removing "invalid slopes" instead of using slope replacement as we did in Fig. 3 (see Supplementary Methods, Preprocessing of the slope). As shown in Supplementary Fig. 2, cTBS effects by left dLMC stimulation were still significant for both tone and consonant perception in noise, even after removing "invalid slopes" (Fig. 3e and Supplementary Fig. 2g). Nevertheless, since inducing "invalid curve fitting" can also be regarded as a TMS disruption effect (as demonstrated by the fact that TMS induced more invalid slopes in noisy conditions compared to sham, Supplementary Fig. 2e, f), we used the results after the preprocessing of slope replacement.

Overall, these results reveal effector-specific motor engagement in speech perception. Specifically, the dLMC is involved in the perception of both lexical tone and voicing of consonants, while the TMC is selectively involved in dental consonant perception. Our findings thus support the motor hypothesis that the perception of speech is closely linked to the motor somatotopy involved in their production.

## The dLMC contributes to various stages of perceptual decision-making

Next, we used HDDM[20] to determine whether the dLMC is not only involved in the perception but also in the decision-making process, and if it does, what cognitive stages of the decision-making process it may participate in (the temporal question). Drift-diffusion model is a type of sequential sampling model that captures trial-wise variations in speed-accuracy tradeoff[21]. It assumes that perceptual evidence is noisily accumulated over time starting from the point of response bias ($z$), with an average drift rate ($v$), and triggers a decision until one of the two boundaries (with a distance of threshold $a$) is reached (Fig. 1i). HDDM allows simultaneously estimating subject-level parameters and group distributions, and provides posterior distributions of estimated parameters. Here, we modeled data in Experiment 2 and determined which parameters ($a$, $v$, or $z$) were affected by TBS (Fig. 1j–l) by comparing 8 linear regression models recruiting none (baseline), one, two, or all (full) of the three parameters, respectively (see Methods, HDDM and reaction time analysis).

iTBS effects were indistinct as models with parameters added did not consistently outperform the baseline models (Supplementary Fig. 5a–h), but the full models in cTBS conditions kept surpassing the baseline models (Supplementary Fig. 5i–p). We, therefore, focused on cTBS effects. Since the full models in cTBS conditions had acceptable goodness-of-fit (RT distribution of the simulated data had a good similarity to that of the real data in the 95% credible criteria, see Methods, Hierarchical drift-diffusion model and reaction time analysis), we chose them as winning models and assumed that cTBS exerted effects on all three parameters. Hypotheses were tested by comparing posterior distributions of full model parameters in each condition with zero. As shown in Fig. 4a–d and i–l, compared with sham, cTBS significantly broadened the thresholds of decision boundary ($a$) for all the conditions (all posterior distributions $p_{fdr} < 0.05$, two-tailed) except the condition of tone perception in quiet with left dLMC stimulation. The drift rates of evidence accumulation ($v$) were decreased by left dLMC stimulation in the tone-in-quiet task ($p_{fdr} = 0.014$, two-tailed, Fig. 4a), and were increased by left dLMC stimulation in all remaining conditions (all $p_{fdr} < 0.01$, two-tailed, Fig. 4b, i, j), but were not affected by right dLMC stimulation (Fig. 4c, d, k, l). In contrast, the starting points of evidence accumulation (response bias, $z$) were only affected for consonant in noise by both left ($p_{fdr} < 0.001$, two-tailed, Fig. 4f) and right ($p_{fdr} = 0.012$, two-tailed, Fig. 4h) dLMC stimulation.

We then compared cTBS effects on HDDM parameters and those on RTs. Using linear mixed-effects (lme) models, we found that cTBS significantly increased RTs (effects of cTBS vs. sham on the random intercept of the lme models, $ps_{fdr} < 0.05$) except for the tone-in-noise condition when the left dLMC was stimulated (Fig. 4 e–h, m–p). This corresponds to the general increase in thresholds $a$ as the decision boundary is closely related to the strategy of speed-accuracy tradeoff[28]. For RT analyses in iTBS conditions, see Supplementary Fig. 6.

Meanwhile, the number of HDDM parameters being altered by cTBS predicted the significance of the modulation on slopes of psychometric curves. At least two parameters ($a + v$ and/or $z$) were significantly modulated by cTBS in conditions with significant slope reduction (Fig. 4b, i, j, l, labeled in purple), whereas only one parameter ($a$ or $v$) was modulated by cTBS in conditions without slope alteration (Fig. 4a, c, d, k, labeled in gray). Hence, clear consistency exists between the two independent analysis pipelines. This increases the confidence that we detected authentic TMS effects instead of artifacts induced by psychometric curve fitting and HDDM modeling procedures (patterns generated by data processing would hardly be replicated in two completely different pipelines).

## The dLMC engagement depends on the hemisphere and task difficulty

Lastly, to further uncover the hemispheric asymmetry of the dLMC involvement and how it is modulated by task difficulty (the second spatial question), we integrated results from psychometric curve-fittings and those from HDDM modeling in Experiment 2. We focused on revealing systematic differences among indices (i.e., psychometric slope, and HDDM parameters $a$, $v$, and $z$) between conditions with left/right dLMC stimulation and with/without noise masking. A quantitative summary of the results is shown in Table 1.

We found that the dLMC involvement was left-dominant for both tone and consonant perception. For slopes, tone perception was only hampered by cTBS on the left dLMC and with noise masking (Fig. 3e); consonant perception was inhibited by cTBS on the left dLMC regardless of noise masking (Fig. 3c, e), but was only affected by the right dLMC stimulation when noise was presented (Fig. 3f). For HDDM parameters, cTBS affected drift rates ($v$) in all conditions with the left dLMC stimulated, but not in conditions with the right dLMC stimulated (Fig. 4). In sum, the left dLMC weighs higher than its right counterpart, as cTBS on the left dLMC induced greater changes in the perceptual decision. However, cTBS on both left and right dLMC lengthened RTs and widened thresholds for decision boundary equivalently (Fig. 4), suggesting that the right dLMC also makes contributions.

Masking noise increased the task difficulty and the engagement of the right dLMC. For slopes, only consonant perception in quiet was affected by the left dLMC stimulation (Fig. 3c), but perception in noise was more susceptible to cTBS on either the left or right dLMC (Fig. 3e, f). Note that, the right dLMC stimulation affected the slope of consonant perception in noise (Fig. 3f), indicating a causal role of the right dLMC in challenging listening conditions. In other words, the dLMC involvement in speech perception shifts from left-lateralized to bilateral as cognitive demands increase, supporting the redundancy hypothesis that the right dLMC (or, more generally, right motor cortex) is redundant for this task and offers compensation in adverse listening contexts.

Another factor in our study that affected the task difficulty was the ambiguity of stimuli (i.e., the step in a continuum), as stimuli with pitch contour (for tone) or VOT (for consonant) closer to categorical boundaries are more indistinguishable. We focused on the HDDM results for Experiment 2 and sought to determine whether interactions existed between stimulus ambiguity and cTBS effects on each parameter. We compared full models (cTBS affected $a$, $v$, and $z$) without interaction terms (baseline) and those with one, two, or all three (full) parameters interacting with the stimulus ambiguity (see the Supplementary Methods, Interactions between cTBS effects and stimulus ambiguity for details). We found that across all cTBS conditions, the full models outperformed the baseline models and most of the remaining models. Simple main effect analyses for conditions with significant slope modulation showed that cTBS only affected the boundary ($a$) and the drift rate ($v$) when the stimuli were unambiguous (steps 1 and 5) or half-ambiguous (steps 2 and 4) (Supplementary Table 3). In other words, the dLMC may not be engaged in speech perceptual decisions when acoustic information is categorically ambiguous.

## Discussion

This study investigates the role of bilateral dLMC in speech perceptual decisions and aims to answer three key questions. The first question addresses whether the dLMC is recruited in speech perception in an effector-specific manner. Results from Experiment 1 and 2 converged to demonstrate that the involvement of the dLMC in speech perceptual decisions is indeed effector-specific. Experiment 1 showed that rTMS on the dLMC, but not the TMC, inhibited both lexical tone and consonant perception as measured by the slope of the psychometric function. Experiment 2 further supported this finding by demonstrating that cTBS

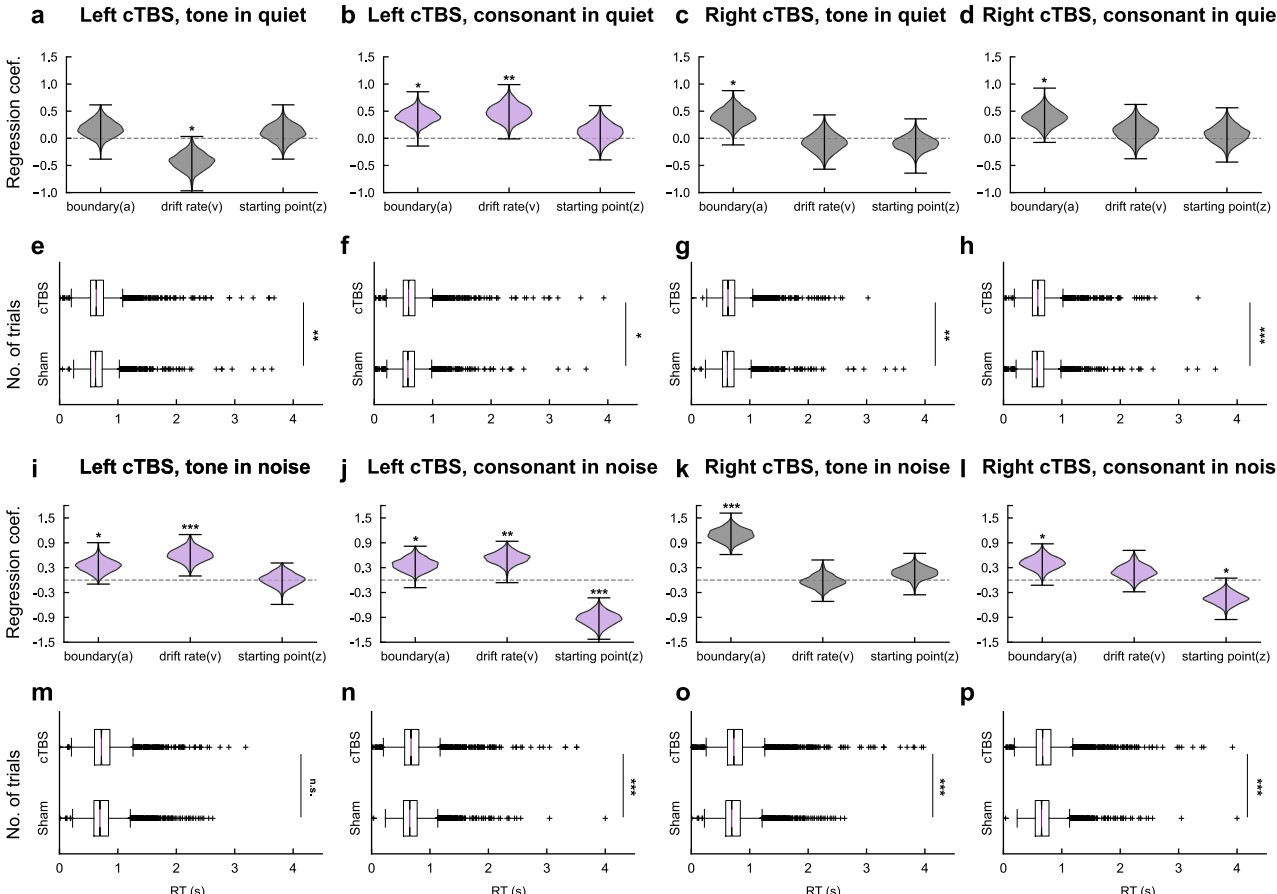

**Fig. 4 | cTBS effects on HDDM parameters and RTs in Experiment 2. a, e** cTBS on the left dLMC decreased drift rates ($p_{fdr} = 0.014$, **a**), and lengthened RTs ($p_{fdr} = 0.002$, **e**) in tone perception in quiet. **b, f** cTBS on the left dLMC increased thresholds of boundary ($p_{fdr} = 0.010$, **b**) and drift rates ($p_{fdr} = 0.005$, **b**), and lengthened RTs ($p_{fdr} = 0.017$, **f**) in consonant perception in quiet. **c, g** cTBS on the right dLMC increased thresholds of boundary ($p_{fdr} = 0.010$, **c**), and lengthened RTs ($p_{fdr} = 0.007$, **g**) in tone perception in quiet. **d, h** cTBS on the right dLMC increased thresholds of boundary ($p_{fdr} = 0.010$, **d**), and lengthened RTs ($p_{fdr} < 0.001$, **h**) in consonant perception in quiet. **i, m** cTBS on the left dLMC increased thresholds of boundary ($p_{fdr} = 0.020$, **i**) and drift rates ($p_{fdr} < 0.001$, **i**), but had no significant effects on RTs ($p_{fdr} > 0.05$, **m**) in tone perception in noise. **j, n** cTBS on the left dLMC increased thresholds of boundary ($p_{fdr} = 0.011$, **j**), drift rates ($p_{fdr} = 0.005$, **j**), and altered starting points ($p_{fdr} < 0.001$, **j**), and lengthened RTs ($p_{fdr} < 0.001$, **n**) in consonant perception in noise. **k, o** cTBS on the right dLMC increased thresholds of boundary ($p_{fdr} < 0.001$, **k**), and lengthened RTs ($p_{fdr} < 0.001$, **o**) in tone perception in noise. **l, p** cTBS on the right dLMC increased thresholds of boundary ($p_{fdr} = 0.011$, **l**) and altered starting points ($p_{fdr} = 0.012$, **l**), and lengthened RTs ($p_{fdr} < 0.001$, **p**) in consonant perception in noise. For HDDM parameters (**a**–**d** and **i**–**l**), each violin

plot demonstrates the normalized distribution of HDDM simulated samples ($n = 1980$) derived from 3900 trials from 25 participants; whiskers represent the lower and upper extremes of the distribution. For reaction time analyses, **e**–**h** and **m**–**p** show the reaction time distributions (3900 trials from 25 participants in each condition). Box-whisker plots demonstrate the entire distributions (center line: median; limits of boxes: 25th and 75th percentile (Q1 and Q3); whiskers: Q1 − 1.5 interquartile range (IQR) and Q3 + 1.5*IQR; points: outliers). For HDDM analyses (**a**–**d** and **i**–**l**), full HDDM models assuming that cTBS affected all three parameters ($a$, $v$, and $z$) were selected and posterior distributions of parameters compared with sham were shown. $P$ value was defined by the probability that the posterior distribution is above or below zero (two-tailed). Note that, conditions with slopes significantly impaired by cTBS (see Fig. 3) were shown in purple, whereas the other conditions were shown in gray. For reaction time analyses (**e**–**h** and **m**–**p**), single-trial RTs were compared between cTBS and sham (two-tailed) using the lme with fixed slope (cTBS effect) and random intercept (participant). For all statistical tests, $P$ values were adjusted by false discovery rate (FDR) correction (threshold = 0.05). *$p_{fdr} < 0.05$; **$p_{fdr} < 0.01$; ***$p_{fdr} < 0.001$. Source data are provided as a Source Data file.

on either the left or right dLMC modulated tone and consonant perception, as indexed by changes in psychometric slope and/or HDDM parameters. The second question addresses how bilateral dLMC work together under different task difficulties. Experiment 2 revealed that the dLMC engagement was bilateral, with the left hemisphere showing dominance and the right counterpart upregulated in cognitively demanding tasks. Finally, the third question concerns what stages of the decision-making process the dLMC is engaged in. Experiment 2 showed that bilateral dLMC contributed to all stages of speech perceptual decision, including the starting point (response bias), drift rate, and boundary threshold, with specific parameters depending on the stimulated hemisphere and task difficulty. Since Experiment 1 was exploratory, we interpret our results in the following discussion based solely on the findings from Experiment 2.

Our findings support the motor hypothesis (Fig. 1d), as evidenced by the recruitment of bilateral dLMC with effector-specificity. The dLMC is known to regulate vocal pitch and control voicing during speech and singing[10,29] by adjusting the tension and adduction/abduction of the vocal folds in the larynx[15], allowing speakers to produce pitch contour and voicing contrasts, respectively. Here, we demonstrated that bilateral dLMC were involved in the perception of both speech features, indicating that they may simulate pitch control and voicing functions accordingly to aid perception. Previous TMS studies have reported that the dLMC is causally involved in the discrimination of non-lexical vocal pitch[9,30] and singing voice[31]. A recent study demonstrated that lexical tone processing is suppressed after stimulating a lip region close to the dLMC in the left motor cortex[32]. The current study takes a step further and establishes a causal link

**Table 1 | Modulations of cTBS effects by hemisphere and task difficulty in Experiment 2**

| cTBS target | Indices | In quiet (Figs. 3, 4) | | In noise (Figs. 3, 4) | | Completely ambiguous (Supplementary Table 3) | |
|---|---|---|---|---|---|---|---|
| | | Tone | Cons. | Tone | Cons. | Tone | Cons. |
| **Left dLMC** | Slope | - | + | + | + | n.a. | n.a. |
| | RT | + | + | - | + | n.a. | n.a. |
| | *a* | - | + | + | + | - | - |
| | *v* | + | + | + | + | - | - |
| | *z* | - | - | - | + | + | - |
| **Right dLMC** | Slope | - | - | - | + | n.a. | n.a. |
| | RT | + | + | + | + | n.a. | n.a. |
| | *a* | + | + | + | + | n.a. | - |
| | *v* | - | - | - | - | n.a. | - |
| | *z* | - | - | - | + | n.a. | n.a. |

Results are adopted from Fig. 3 (cTBS effects on slopes), Fig. 4 (cTBS effects on HDDM parameters), and Supplementary Table 3 (interactions between cTBS effects on HDDM parameters and stimulus ambiguity).

Completely ambiguous: trials where the tone or consonant was at the categorical boundary (i.e., step 3) of the continuum and completely ambiguous.

+: significant cTBS modulation effect; −: no significant cTBS effect; n.a.: not analyzed.

*dLMC* dorsal laryngeal motor cortex, *Cons.* consonant, *RT* reaction time, *a* boundary threshold, *v*- drift rate; *z* starting point.

between the human dLMC (defined by BOLD activations in articulation tasks) and perceptual decision of lexical tone. In addition, we provide previously unreported evidence that bilateral dLMC were causally recruited in detecting subtle VOT differences for plosive consonants. This parallels the dLMC's motor function to accurately control the moment of voicing onset. Notably, the consonant results are inconsistent with ref. 9 as they found that stimulating the dLMC on neither side affected consonant perception. However, our tasks were more challenging due to extra noise masking, which might urge the dLMC engagement. Overall, previous studies have reported that the right motor cortex entrains the speech stream[33] and is involved in auditory-speech coupling[34] resembling its left counterpart, and we provide new evidence for its engagement in the perception of lower-level phonemic cues.

Our findings also support the redundancy hypothesis (Fig. 1e) by demonstrating left asymmetry and right upregulation in difficult tasks for the dLMC engagement in speech perception. The redundancy hypothesis is based on information theory and highlights the degeneracy and redundancy of cognitive anatomy, i.e., many-to-one brain structure-function relationship and multiple structures encoding the same information in parallel[35]. Such properties facilitate stable and flexible functional adaptations of organisms[36], which is particularly beneficial when the computational demands increase due to brain injury or overload. Upon such challenges, other nodes within the same network, including contralateral homologous regions, may be upregulated and compensate for cognitive impairment via functional reorganization[37]. Particularly, for language tasks that engage the right hemisphere, ref. 38 found impaired speech comprehension in subjects with right hemisphere lesion, while ref. 39 found that the right hemisphere gray matter volume predicts language ability in individuals with chronic stroke-induced aphasia. Moreover, ref. 40 used cTBS to inhibit the left inferior frontal gyrus in subjects with left temporoparietal lesions and found upregulation in the right lesion homologous to compensate for weakened phonological processing. Hence, the right hemisphere may parallel the left homolog and is reorganized to be a key node as challenges increase during language processing. Here, we provide further evidence for such compensatory functional redistributions by showing the upregulation of the right motor cortex in adverse speech perception tasks. Nevertheless, in addition to

contralateral recruitment, functional reorganization may also be supported by the upregulation of domain-general networks[37] and fluctuate at different phases of brain plasticity[41]. Taking these into account, future research needs to integrate brain imaging techniques to explore the whole-brain dynamic reorganization in speech perception as a function of cognitive demand.

In contrast, our results only provide weak support for the lexical hypothesis, which suggests that the functional asymmetry of the motor cortex is due to differences in linguistic functions between lexical and non-lexical speech cues. This hypothesis contrasts with the acoustic hypothesis, which proposes that differences in stimulus acoustic properties (i.e., temporal and spectral modulation) drive the asymmetry[11]. Previous studies have shown that the motor engagement for lexical and segmental units is left-dominant[2,32,42], whereas that for the non-lexical and suprasegmental prosody is biased to the right hemisphere[9,32]. Tang et al. (2021)[32] used the oddball paradigm and found that linguistic functions drove the asymmetric auditory-motor processing of lexical tone in tonal and non-tonal language speakers, and the pre-attentive detection of lexical tone changes was left-lateralized in tonal language speakers. Using an explicit identification paradigm, our study supports the lexical hypothesis but not the acoustic hypothesis by showing that the motor engagement for both lexical tone and consonant perception were left-biased. However, our evidence was weak as the observed hemispheric asymmetry was more sophisticated and modulated by task difficulty. Thus, the right motor cortex may be redundant instead of functionally specialized for non-lexical speech cues. Furthermore, contested results exist in the literature, as some studies found left motor activations during lexical tone perception in both tonal and non-tonal language speakers[43], but some found bilateral motor engagement during lexical tone perception with enhanced right auditory-motor connectivity[44]. Overall, our findings add to the complex understanding of the neural mechanisms underlying motor asymmetry and suggest that further research is needed to fully elucidate this phenomenon.

Consistent with the interactive model of auditory-motor speech perception proposed by ref. 19, we found that the dLMC contributes to various stages of perceptual decision-making, but depending on the hemisphere and task difficulty. The effects of dLMC stimulation on RTs and HDDM boundary threshold *a* were consistent across most conditions, regardless of the hemisphere and task difficulty. However, the drift rate *v* was only affected by left dLMC stimulation, but not by right stimulation. In addition, the response bias (i.e., starting point *z*) was influenced by cTBS on both the left and the right dLMC only in difficult tasks (i.e., consonant perception in noise). Notably, the observed TMS effects on speech perception remain a matter of debate, as some studies suggest that they may only reflect changes in response bias rather than motor involvement in speech perception[45]. Our study provides a more nuanced perspective, suggesting that bilateral motor engagement may be associated with response bias when perception is cognitively demanding. However, the relationship between the remaining two HDDM parameters (i.e., boundary threshold *a* and drift rate *v*) and the two underlying perceptual stages, namely acoustic-phonetic feature extraction and phonemic category mapping, remains elusive. Previous studies have postulated that the boundary threshold is related to the ambiguity of sensory input[28,46], while the drift rate is influenced by both the stimuli saliency[46] and top-down predictions derived from frontal regions[28]. One possible interpretation of our findings is that bilateral dLMC provides an articulatory efference copy[47] that serves as categorical templates to match with sensory input. Categorical templates increase the perceptual distance (i.e., sensitivity) between signals by non-linearly transforming the psychological space[26]. Hence, when the dLMC was blocked, perception became more uncertain. The faster drift rates after left dLMC stimulation may reflect compensatory processes originating from a left-lateralized, knowledge-based frontal language system[48] that uses prior

knowledge to offset losses of motor templates in perceptual decision. These compensatory processes were particularly evident (i.e., more HDDM parameters were modulated) when the motor template losses were sufficient to affect perceptual sensitivity (i.e., the slope was also lowered).

Limitations exist in the current study regarding the location and function of the dLMC. Firstly, the dLMC stimulation targets we defined (z = 50) were superior to the activation peak of the dorsal larynx/phonation area (LPA, corresponding to the dLMC) found by Brown et al. (2018) (z = 44)[13]. However, while we selected the dLMC targets based on brain activations that recruited laryngeal functions, the previously found LPA is within the ventral-dorsal range of activations for vowel production in the current study and the effective radius (1.5 cm) of TMS[25] (See Supplementary Fig. 4). In addition, research has found two subregions in the human motor cortex to control laryngeal movements: the ventral and dorsal LMC[14], belonging to separate laryngeal movement control networks[49]. The vLMC is a homologous area of the non-human primate LMC, while the dorsal region may have evolved as a human-specific area[50]. Although the functions of these two LMCs are not fully understood, the dLMC may be closely associated with complex human language and singing abilities[10]. Nevertheless, future study should still compare the functions of the two LMCs in speech perception. Also, future studies are needed to test the effects of tonal language experience on the location of the dLMC, as previous localization work has been biased towards non-tonal language users. Secondly, while we demonstrated that the dLMC was recruited in distinguishing lexical pitch contours (level vs. rising tone), it is also possible that the dLMC encodes merely the mean F0 value that also differentiates the two tones (see Supplementary Table 1). Although this does not affect our main conclusions, it is an interesting topic for future studies to dissociate the dLMC functions for pitch height and pitch contour perception.

As for the TMS methodology, we only found the "virtual lesion" effects by cTBS but not any excitatory iTBS effects. It is possible that the motor cortices were already fully recruited and resistant to further enhancement. Indeed, most previous TMS studies have only reported inhibitory effects on motor engagement in speech perception (e.g., refs. 9,51), suggesting that speech perception improvement through TMS on the motor cortex may be challenging. Further studies exploring cognitive enhancement through TMS are necessary for clinical applications. Also, our work is unable to demonstrate the neurobiological infrastructure for causal inferences for neurostimulation. Short-term functional reorganization of neural networks may occur after stimulation, complicating the interpretation of causality[52].

Additionally, the perception of sentences with rhythmical, hierarchical, and contextual information may necessitate more sophisticated functions of the motor cortex, such as temporal[53] and contextual[54] prediction, as well as envelope tracking[55], in addition to the perception of phonemic cues. To fully understand the functional importance of dLMC in speech perception in natural settings, future research is required.

In conclusion, as illustrated in Fig. 5, we propose a model of bilateral dLMC engagement in the temporal dynamics of perceptual decision for lexical speech cues. Firstly, the recruitment of bilateral dLMC for the perceptual decision of Mandarin lexical tone and voicing of consonant confirms the motor hypothesis regarding bilateral effector-specificity. Secondly, the left dominance of the dLMC and the compensatory engagement of its right counterpart under increased difficulty support the redundancy hypothesis. Thirdly, bilateral dLMC are involved in multiple stages of the perceptual decision-making process, which is mediated by the hemisphere and cognitive demands. This study provides strong support for the sensorimotor integration account of speech perception[2,3]. Moreover, our findings broaden the scope of research on this topic by providing insight into the functional spatial distributions of bilateral motor cortices and their involvement in the temporal dynamics of speech perceptual decision-making.

## Methods

### Participants

Participants were graduate students or employees in Beijing with higher education, and were recruited through advertisement (convenience sampling). Sixty-four young participants (36 females, mean age = 22.19 years, SD = 2.88) took part in Experiment 1 (one left before finishing the sham and dLMC stimulation condition, leaving 63 participants with valid data). Participants were divided into two matched groups for left and right hemisphere stimulation, respectively. For each group, the group size (left: 35; right: 28) is sufficient to detect medium to large effects as estimated by G*Power 3.1. In each group, 2 participants did not finish the TMC stimulation sessions, leaving the sample sizes in TMC conditions smaller (left: 33; right: 26) but still sufficient. Another 26 participants (14 females, Mean age = 22.38 years, SD = 2.88) took part in Experiment 2 (one stopped after one session with iTBS upon the right dLMC, leaving 25 participants with valid data). The group size is sufficient to detect medium to large effects as estimated by G*Power 3.1. All participants were right-handed as assessed by Edinburg Handedness Inventory (mean laterality quotient: Experiment 1: 81.45, SD = 20.71; Experiment 2: 79.60, SD = 18.86)[56], and had normal hearing as measured by the Bell Plus audiometer (Bell, Inventis, Italy) with a TDH-39 headphone at 0.125, 0.25, 0.5, 1, 2, 4, and 8 kHz (≤20 dB SPL). None of them had self or family history of neurological, traumatic, or psychiatric diseases. Participants were Mandarin Chinese speakers, and were all non-musicians (no more than 2 years of music training) as assessed by the Chinese version of Montreal Music History Questionnaire (MMHQ)[57]. All participants gave written informed consent prior to the experiment. They were paid ￥50 per hour for participation in behavioral sessions, and ￥100 per hour for TMS sessions. The experiment was approved by the Ethics Committee of the Institute of Psychology, Chinese Academy of Sciences.

### Stimuli generation

Syllables were recorded by a young female Mandarin speaker (age = 24). Sounds were recorded in a sound-proof studio, by RODE NT1 microphone connected to an Audient ID14 soundcard (sampling rate = 44.1 kHz). Syllables were [ti55], [ti35], [thi55], and [thi35] (transcribed by the International Phonetic Alphabets[58] and Chao's system of "tone-letters"[59]), corresponding to Mandarin syllables "堤dī" (riverbank), "敌dí" (enemy), "梯tī" (ladder), and "题tí" (title) (marked by Chinese Phonetic Alphabets "Pinyin"[60]), respectively (Supplementary Audio 1). Syllables were also presented to participants in Pinyin for task instructions (all participants have received Pinyin education). Here, Tone1 (55) and Tone2 (35) refer to the high-level tone and the high-rising tone, respectively; "d" and "t" refer to the dental plosive [t] with shorter VOT (unaspirated) and the dental plosive [th] with longer VOT (aspirated), respectively. Four syllables were 5 kHz low-passed filtered by Praat 6 algorithms and matched according to the average root mean square sound pressure level (SPL). To control acoustic features, four syllables were noise-removed, and re-concatenated, and their pitch contours were normalized by Praat 6 (see Supplementary Table 1 for acoustic properties of the syllables).

For Experiment 1, based on four natural syllables, 5 × 5 -step tone (Tone1 to Tone2) and consonant ([t] to [th]) continuum matrices with individualized ranges of ambiguity were generated by stepwise adjusting F0 and appending aspiration noise (Fig. 1b, left panel). Continua were synthesized by Matlab R2016a and legacy-STRAIGHT (2018). For Experiment 2, individualized five-step tone ([ti55] to [ti35], Supplementary Audio 2) and consonant ([ti55] to [thi55], Supplementary Audio 3) continua were generated in similar ways (Fig. 1c, left panel). To avoid interactions between tone and consonant perception, no

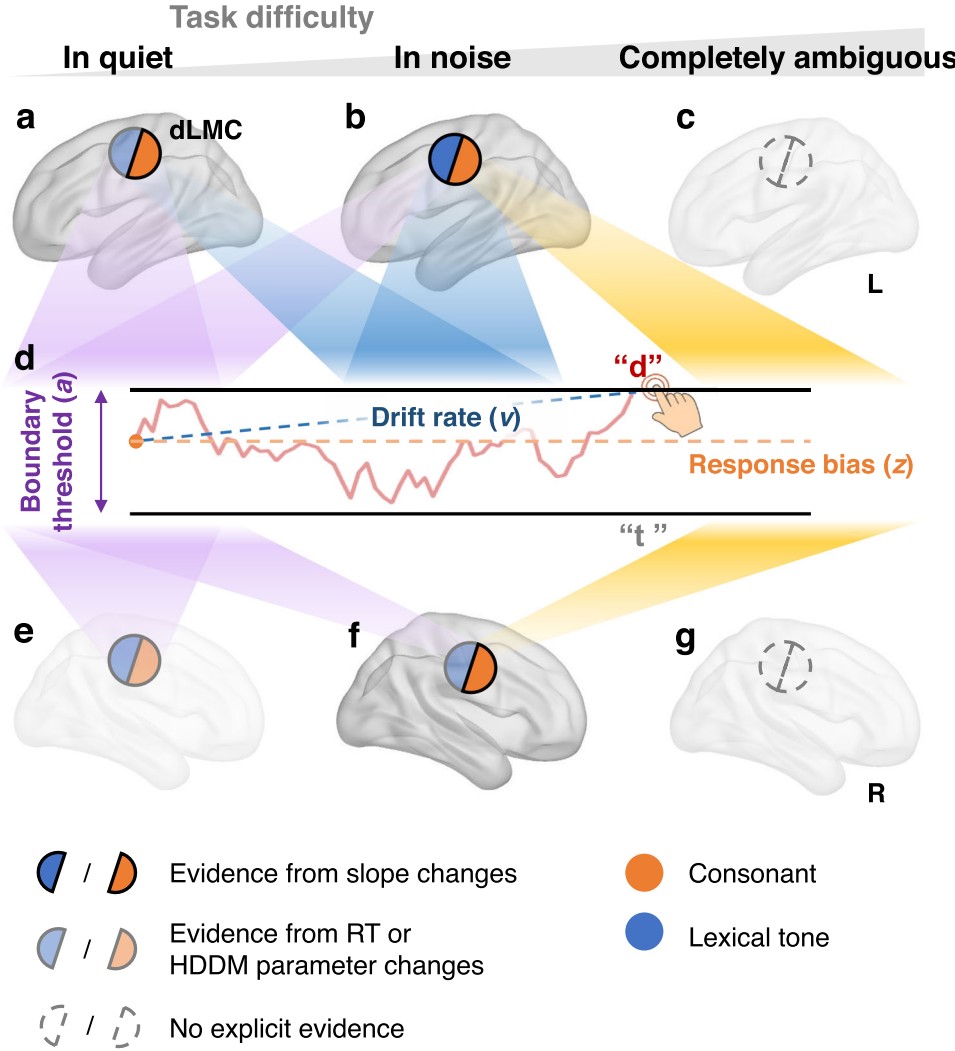

**Fig. 5 | The proposed model of bilateral dLMC engagement in the perceptual decision of speech cues.** Bilateral human dLMC are engaged in multiple stages of the perceptual decision process of Mandarin lexical tone and voicing of consonant as a function of task difficulty. In the spatial domain, this supports the motor hypothesis (with bilateral effector-specific engagement) and the redundancy hypothesis (with left dominance and right upregulation as difficulty increases). In the temporal domain, this reveals the recruitment of the dLMC in multiple steps of perceptual decision, which is also modulated by the hemisphere and task demands. **a–c** The involvement of the left dLMC in quiet (**a**), in noise (**b**), and when the speech stimulus is completely ambiguous (**c**). **d** Perceptual decision process depicted by hierarchical drift-diffusion model (HDDM) with three determining parameters: boundary threshold $a$, drift rate $v$, and starting point or response bias $z$. **e–g** The involvement of the right dLMC in quiet (**a**), in noise (**b**), and when the speech stimulus is completely ambiguous (**g**). Semicircles in (**a–c**) and (**e–g**) demonstrate evidence for the effector-specific engagement (whether or not engaged in tone and/or consonant perception) of the left/right dLMC at each difficulty level, as revealed by cTBS modulation. Opaque semicircles: cTBS changed the psychometric slope (perceptual sensitivity); semitransparent semicircles: cTBS only changed the RTs or DDM parameters; transparent semicircles: no cTBS effects were found. orange: consonant tasks; blue: lexical tone tasks. Spotlights from (**a–c**) and (**e–g**) to (**d**): engagement of the dLMC in affecting the stages (parameters) of the perceptual decision at different difficulty levels. Purple: boundary threshold $a$; blue: drift rate $v$; orange: response bias $z$. BrainNet Viewer was used to generate the schematic brain maps in (**a–c**) and (**e–g**). COPYRIGHT NOTICE: © Copyright 2007, NITRC. All rights reserved. [https://www.nitrc.org/include/copyright.php] This illustration is based only on cTBS results from Experiment 2. See Table 1 for a quantitative summary.

continuum matrix was used in Experiment 2. See Supplementary information, Continua generation and Supplementary Table 1 for details.

**Stimuli presentation**
Stimuli were played by Sennheiser in-ear headphones (IE 60) connected to a Steinberg UR 242 amplifier. Intensity levels of stimuli were calibrated by Larson-Davis sound level meter (Model 831, Depew, NY). In Experiment 1, syllables were presented by Psychtoolbox-3 in Matlab 2016a at 78 dB sound pressure level (SPL), loud enough to resist online TMS noise. In Experiment 2, syllables were presented at 65 dB SPL. This was lower than that in Experiment 1 since no TMS noise affected stimuli presentation.

In conditions with noise masking in Experiment 1, speech-spectrum noise (SSN) started as a block started and was kept playing until the block ended. SSN was generated by modulating white noise with the average spectrum of 5000 sentences spoken by ten young female Mandarin speakers[61], and 4 kHz low-pass filtered. During the presentation, the intensity of masking noise was adjusted to produce a signal-to-noise ratio (SNR) of −4 dB, a medium level that was shown to induce maximal sensorimotor integration in syllable decision[62,63]. In Experiment 2, we applied SSN identical to Experiment 1 (except 5 kHz low-pass filtered) in conditions with noise masking. The intensity of SSN was adjusted to produce individualized SNRs. In each trial, SSN began and ended simultaneously with the syllable, with additional 10 ms linear rise-decay envelopes.

## MRI acquisition and processing

For each participant in both experiments, a T1-weighted anatomical image for target mapping was acquired by a 3-Tesla Siemens Magnetom Trio scanner with a 20-channel head coil, using the magnetization-prepared rapid acquisition gradient echo (MPRAGE) sequence (TR = 2200 ms, TE = 3.49 ms, field of view = 256 mm, flip angle = 8°, spatial resolution = $1 \times 1 \times 1$ mm). DICOM images were converted to NIfTI format by dcm2nii in MRIcron (2016).

## Experiment procedure

**Experiment 1.** Participants finished Experiment 1 on five separate days: day 1 for filling questionaires, assessing hearing and TMS thresholds, and preliminary experiments to estimate personal ranges of perceptual ambiguity of tone and consonant perception; day 2 for MRI acquisition; day 3 to 5 for rTMS experiments. rTMS experiments included three sessions: one sham session, one dLMC stimulation session, and one TMC stimulation session. The dLMC and TMC targets were defined by the group-level MNI coordinates of geometric middle points of BOLD activations in a functional MRI localization experiment, where 48 participants performed larynx-related (say [a]) and tongue-related (say [tʰ]) articulation tasks (dLMC: [±40, −5, 50], TMC: [±59, −3, 36]; see Supplementary Methods, Functional localization experiment). Each of the three sessions was conducted separately for at least 3 days. The order of sessions was counterbalanced across participants using a balanced Latin square procedure.

Within each session, participants underwent four blocks of syllable identification tasks: tone/consonant tasks with/without noise masking. All four blocks used the same individualized $5 \times 5$-step matrix. The order of four blocks was counterbalanced across participants in a Latin square procedure. One block contained two mini-blocks. Each of the mini-blocks randomly presented all syllables in the $5 \times 5$-step tone–consonant matrix. To enhance the reliability of the fitted slope changes, in each mini-block, additional trials around the PSE were presented. This yielded a total of 78 trials, with 39 trials in each mini-block. In each trial, participants heard a syllable while receiving online rTMS. After a syllable was presented, participants used their index and middle finger to press the "<" or ">" keyboard key to make a speeded judgment (to which tone/consonant category the heard syllable belonged), and ignore the other dimension. Participants always used the hand ipsilateral to the stimulated hemisphere to avoid response bias induced by stimulating the hand motor cortex. The order of keys was balanced across participants. In each trial, a resting period of maximal 5 seconds after sound onset was left for a response. The next trial started 2 s on average (1.8–2.2 s, 10 ms step, uniformly distributed) after participants' response or the automatic end of a trial. No break time was left between adjacent mini-blocks.

To get prepared for the experiment, participants completed practice tasks before each session of the rTMS experiment. The practice task preceding the first rTMS session contained short tone/consonant identification tasks using a different continuum matrix ([p]–[pʰ] × Tone1–Tone2 in Mandarin), whereas the practice tasks before the second and third sessions included a short version of the formal experiment.

**Experiment 2.** Participants finished Experiment 2 on seven separate days. Procedures for the first and second days were similar to Experiment 1. Day 3 to 7 were for TBS experiments, which included five sessions: sham, iTBS on the left dLMC, cTBS on the left dLMC, iTBS on the right dLMC, and cTBS on the right dLMC. Each of the five sessions were conducted separately for at least 3 days. The order of sessions was counterbalanced across participants using a balanced Latin square procedure.

Within each session, participants first received one train of TBS or sham stimulation. After stimulation, participants underwent four blocks of syllable identification tasks similar to Experiment 1.

Individualized tone and consonant continua were used for tone and consonant tasks, respectively. The order of four blocks was counterbalanced across participants in a Latin square procedure. One block contained four mini-blocks. Each of the mini-blocks randomly presented all syllables in the continuum. To enhance the reliability of the fitted slope changes, in each mini-block, additional trials around the PSE were presented. This yielded a total of 156 trials, with 39 trials in each mini-block. Participants responded in the same way as in Experiment 1. In each trial, a resting period of maximal 5 s after sound onset was left for a response. The next trial started at 0.5 s on average (0.4–0.6 s, 10 ms step, uniformly distributed) after participants' response or the automatic end of the trial. No break time was left between adjacent mini-blocks. Identical to Experiment 1, participants completed practice tasks before each TBS session.

## TMS

**TMS resting motor threshold estimation (Experiment 1).** Before the experiment, participants received two to three pulses of TMS at middle-intensity levels to guarantee tolerance. Participants' resting hand motor threshold (RMT) was tested by placing the coil on the dorsal motor cortex localized by Brainsight 2.4 neuronavigation software and adjusting output intensity stepwise. The RMT was defined as the lowest stimulator output that induced at least five of ten trials with visible hand motor responses. Participants' average RMT was 61.48% of maximal stimulator output (SD = 7.97%).

**TMS paradigm (Experiment 1).** Participants were separated into two groups for left (N = 35, 20 females) and right (N = 28, 15 females) hemisphere stimulation, respectively. Two groups of participants were matched in age (left: M = 22.43, SD = 3.03; right: M = 21.96, SD = 2.69; $t(60.22) = 0.63$, $p = 0.53$), RMT (left: M = 61.09%, SD = 7.69; right: M = 61.96%, SD = 8.28; $t(61) = -0.43$, $p = 0.67$), and handedness (left: M = 80.26, SD = 23.72; right: M = 82.94, SD = 16.05; $t(61) = -0.50$, $p = 0.62$).

During the experiment, triple-pulse rTMS was applied in synchronization with the syllables. Biphasic magnetic pulses were generated by a standard figure-of-eight 70 mm coil from a Magstim Rapid2 stimulator. The intensity of stimulation was set to 90% of participants' RMT. The frequency of stimulation was 10-Hz (100 ms between two pulses). The first pulse was always delivered at the syllable onset, followed by two subsequent pulses. Since a single TMS pulse induces aftereffects lasting for tens of milliseconds[64], a triple-pulse sequence of stimulation can shadow an entire syllable. Also, although it was suggested that high-frequency offline rTMS induce facilitatory effects[65], 10-Hz online rTMS protocol has been shown to inhibit cognitive processing[66] and, more specifically, downregulate speech perception[67]. Presumably, while offline TMS alters performance by inducing short-term neural plasticity[68], online TMS affects cognition by modulating ongoing neural oscillations[69], implying that the frequency-dependent rule of offline rTMS effects may not fit online paradigms. To account for inter-subject variability in brain structure, MNI coordinates of the dLMC and TMC targets were mapped to the participants' T1-weighted images with Brainsight, and were manually adjusted if coordinates were not in precentral gyrus while keeping the vertical axis positions constant. Positions of the coil were real-time tracked by the Brainsight system to keep the target being focused. The magnetic field was always kept perpendicular to the precentral gyrus, heading towards the front of the participant. In sham conditions, the coil was placed on the vertex and the same stimulation sequences were delivered.

**Active motor threshold estimation (Experiment 2).** Participants' active motor threshold (AMT) was estimated with motor evoked potentials (MEPs) recorded from the right first dorsal interosseous muscle and were automatically filtered by Brainsight 2.4. Here we used

AMT instead of RMT because AMT has been confirmed to be a valid indicator of individualized TBS intensity levels[70]. Note that, we found that, in Experiment 2, participants' RMT and AMT were highly correlated (r = 0.854, $p < 0.001$), showing that AMT captures similar individual differences to what RMT does. Before the experiment, participants accepted two to three pulses of TMS at middle-intensity levels to guarantee tolerance. Participants' RMT was estimated as described in Experiment 1, except that the RMT was defined as the lowest stimulator output that induced at least five of ten trials with MEP peak-to-peak amplitude greater than 50 μV. Based on RMT, output intensity was adjusted to measure AMT. Participants were keeping a maximal voluntary hand contraction to a dynamometer during the measurement, and the AMT was defined as the lowest stimulator output that induced at least five of ten trials with MEP peak-to-peak amplitude greater than 1000 μV. This was similar to the criteria of 200 μV when participants maintained 20% of maximal hand contraction[71], but was easier to control. 80% AMT was used as stimulation intensity during the experiment, but the intensity was always kept at least 40% of the maximal stimulator output.

## Data analyses

**TMS paradigm (Experiment 2).** During the experiment, participants received TBS after practice and before tasks. Biphasic magnetic pulses were generated by a standard figure-of-eight coil from a Magstim Rapid2 stimulator. The intensity of stimulation was set to 80% AMT or 40% of maximal stimulator output. We used the TBS protocol that delivered three 50 Hz pulses in each 200 ms period[71]. For iTBS conditions, TBS was delivered for 2 s followed by an 8 s resting interval. A train of iTBS lasted for 192 s and delivered 600 pulses in total. For cTBS conditions, 600 pulses of stimulation were continuously delivered for 40 s. Navigation of the TMS coil was identical to that in Experiment 1. In sham conditions, the coil was placed over the left or the right dLMC and delivered iTBS or cTBS, but was tilted so that only the edge was kept in touch with the skin. In iTBS conditions, participants started to perform tasks immediately after the stimulation ended. In cTBS conditions, participants performed tasks 5 min after stimulation to maximize TMS aftereffects[71].

**Fitting psychometric function (Experiment 1).** Scores of identification tasks were binarily coded. Logistic models were fitted using the nonlinear *lsqcurvefit* function in Matlab R2016a, taking morphing steps in a continuum as the explanatory variable and scores as the response variable. The formula of the curve is shown below:

$$\hat{\mathbf{Y}} = \frac{a - b}{1 + e^{-\sigma(\mathbf{X} - \mu)}} + b \tag{1}$$

$\hat{\mathbf{Y}}$ is the estimated psychometric function (identification curve). $\mathbf{X}$ is the steps of a continuum. $a$ and $b$ are the upper and lower asymptotes of the curve. $\mu$ is the PSE. $\sigma$ is the slope of the curve at the PSE. Slopes of psychometric functions served as the response variable. The lower the slopes, the weaker the participant's capacity to categorize phonemes.

We used a four-parameter function[72] instead of a more widely used simplified version that fixes the lower and upper asymptotes to 0 and 1. This is because it is unreasonable to assume that participants could perform 100% correctly even when the tone or consonant is unambiguous, especially in the noise masking conditions. Also, to avoid overfitting, for each condition, upper and lower asymptotes ($a$ and $b$) were calculated by an unconstrained fit on the average data, and were fixed for each block. Slopes were then preprocessed to reduce effects from invalid outliers (see Supplementary Methods, Preprocessing of the slope and Supplementary Fig. 1).

**Stimulation effect analyses (Experiment 1).** Unexpectedly, we found competitions between tone and consonant perception such that slopes were affected by the ambiguity of the unattended dimension (see Supplementary Methods, Competitions between consonant and tone perception). Thus, it is necessary to exclude such interference with rTMS effects. Therefore, for each block of tasks per session (dLMC rTMS, TMC rTMS, and sham) per group of participants (left and right hemisphere stimulation), curve fitting, preprocessing, and estimation of rTMS effects were calculated separately for ambiguous (step 3), half-ambiguous (step 2 and 4), and unambiguous (step 1 and 5) trials according to the unattended attention (Supplementary Fig. 3a).

rTMS effects were defined as differences between slopes in rTMS blocks and the corresponding sham blocks (rTMS - Sham) and one-tailed statistical tests were applied with the null hypothesis rTMS - Sham ≥ 0. Also, the relative down-regulation of rTMS upon the dLMC on perception was estimated by comparing dLMC stimulation with TMC stimulation effects on slopes ((dLMC - Sham) - (TMC - Sham)) and one-tailed statistical tests were applied with the null hypothesis (dLMC - Sham) - (TMC - Sham) ≥ 0. Details of statistical estimations are presented in Supplementary Methods, Analysis of TMS effects, and Supplementary Fig. 1.

**Fitting psychometric function (Experiment 2).** Procedures of fitting psychometric function, preprocessing of the slope, and stimulation effect analysis were identical to those in Experiment 1. Nevertheless, trials were not grouped into different categories for curve fitting. In terms of TMS effect comparison, effects of cTBS (cTBS - Sham) and iTBS (iTBS - Sham) were firstly tested against zeros (one-tailed statistical tests, null hypothesis cTMS - Sham ≥ 0 and iTMS - Sham ≤ 0), and then compared with each other by one-tailed statistical tests ((cTBS - Sham) - (iTBS - Sham) ≥ 0).

**HDDM and reaction time analysis (Experiment 2).** Single-trial data (binary decision and reaction times) were entered into HDDM model[21] using the HDDM package (version 0.9.8)[20] based on Python 2.7 to disentangle TBS effects on three key parameters of perceptual decision-making: starting point or response bias ($z$), average drift rate ($v$) of evidence accumulation, and threshold between decision boundaries ($a$). We used HDDM, instead of the equivalently prevailing linear ballistic accumulator (LBA) model that also belongs to the sequential sampling family, for two reasons. First, HDDM assumes only one accumulator that samples evidence and thus better fits our identification paradigm where the two alternatives varied only in one feature (pitch contour or VOT), compared with the LBA with two accumulators. Second, HDDM can be regarded as the signal detection theory with temporal dynamics as it assumes a stochastic process for evidence accumulation. This matches the goals of our psychophysical experiment to detect subjects' ability to segregate acoustic information from ambiguous signals and noise.

First, we built 8 regression models recruiting none (baseline), one, two, or all (full) of the three parameters, respectively. We used dummy treatment coding with the intercept set on sham conditions. Each model drew 2000 posterior samples, and discarded the first 20 samples as burn-in. Models are shown below.

Baseline model: $a$ ~ 1, $v$ ~ 1, $z$ ~ 1.
Model 1 ($a$ only): $a$ ~ TBS, $v$ ~ 1, $z$ ~ 1.
Model 2 ($v$ only): $a$ ~ 1, $v$ ~ TBS, $z$ ~ 1.
Model 3 ($z$ only): $a$ ~ 1, $v$ ~ 1, $z$ ~ TBS.
Model 4 ($a + v$): $a$ ~ TBS, $v$ ~ TBS, $z$ ~ 1.
Model 5 ($a + z$): $a$ ~ TBS, $v$ ~ 1, $z$ ~ TBS.
Model 6 ($v + z$): $a$ ~ 1, $v$ ~ TBS, $z$ ~ TBS.
Model 7 ($a + v + z$): $a$ ~ TBS, $v$ ~ TBS, $z$ ~ TBS.

We used the deviance information criterion (DIC, the lower, the better)[73], a method widely used to assess and compare fit in hierarchical models, to select the winning model. A wining model should

have a DIC value that is (1) lower than the baseline model and (2) consistently lower than most of the competing models in all conditions. For cTBS conditions, model 7 (full model) that assumes cTBS effects on all three parameters was then chosen. For iTBS conditions, however, since no model met the criteria mentioned above, we did not conduct further analyses on the iTBS data (see Supplementary Fig. 5 for comparisons of DIC values across models). Note that, a full model may naturally perform better than the other models merely because it contains more parameters (i.e., overfitting)[20]. Thus, we also simulated the data using the estimated parameters, and compared summary statistics between the real and the simulated data. Results showed that, the similarity fell in the 95% credible criteria for full models in all cTBS conditions.

Second, we tested whether cTBS significantly affected each of the three model parameters in each condition by comparing parameter posterior distributions to zero. Since we set the intercept of regression models on sham conditions, the probability that the posterior distribution is above or below zero (two-tailed) refers to the confidence (the $p$ value) in rejecting the null hypothesis (cTBS exerted no effects). $P$ values were then FDR-corrected for multiple comparisons across all conditions.

Third, we tested whether cTBS significantly altered RTs in each condition as a supplementary analysis for understanding how the dLMC influences strategies of speed-accuracy tradeoff. Since the data were hierarchical as single trials were nested in single participants, we used linear mixed-effects (lme) models to estimate differences in RTs between cTBS and sham conditions. The type of stimulation (cTBS or sham) was entered for the fixed slope, and the participant was entered for the random intercept. Since our goal was to testify whether cTBS effects on slopes existed and to avoid failure of convergence, we did not model random effects on slopes. We used the lme4 package in Rstudio 1.4.1103 (R version 4.0.4) to build the models (RTs -1 + type of stimulation + (1|participant)) and used the lmerTest package to test the models. $P$ values of the fixed slope (cTBS effects) were then FDR-corrected for multiple comparisons across all conditions.

Forth, to determine whether cTBS exerted equivalent effects on HDDM parameters in trials with different levels of perceptual ambiguity (i.e., the acoustic position of the syllable in the consonant or tone continuum), we built HDDM regression models with interaction items between the type of stimulation and the step in the consonant/tone continuum, and conducted simple main effect analyses if interaction existed. See the Supplementary Methods, Interactions between cTBS effects and stimulus ambiguity and Supplementary Fig. 7 for details.

### Visualization
Visualization in all figures was implemented by Matlab R2021a, Microsoft PowerPoint 2019, and Adobe Illustrator 2020. Especially, for brain graphs in Figs. 1, 2, 3, and 5, we used the BrainMesh_ICBM152_smoothed template in BrainNet Viewer 1.7[74] to generate the brain graphs; for brain graphs in Supplementary Fig. 4, visualization was performed by Mango 4.1[75] using TemplateFlow tpl-MNI152NLin6Asym template (https://www.templateflow.org/). We used Matplotlib based on Python 2.7 (https://matplotlib.org/) for illustrating Fig. 4 and Supplementary Figs. 5, 6.

### Reporting summary
Further information on research design is available in the Nature Portfolio Reporting Summary linked to this article.

## Data availability
We provided raw data in Experiments 1 and 2, as well as the bold activation maps for the fMRI pretest. Raw data and instructions for usage have been deposited in the Zenodo database (https://zenodo.org/record/8092010)[76]. Source data are provided with this paper.

## Code availability
We provided custom code for data collection, behavioral data analyses, and visualization in Experiments 1 and 2. Code and instructions for code usage are available at Zenodo (https://zenodo.org/record/8075062)[77].

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

## Acknowledgements

The authors greatly appreciate the excellent work of the technical support staff at the Institutional Center for Shared Technologies and Facilities of the Institute of Psychology, Chinese Academy of Sciences. Also, the authors would like to thank Xiaonan Li and Lei Zhang for their help with neuroimaging data processing, Quan Yuan and Xinyuan Wan for their help with data collection, and Xinyi Qian and Yi Weng for reviewing linguistics and phonetics terms. This work was supported by the STI 2030—Major Projects 2021ZD0201500; the National Natural Science Foundation of China 31822024; the Strategic Priority Research Program of Chinese Academy of Sciences XDB32010300; and the Scientific Foundation of Institute of Psychology, Chinese Academy of Sciences E2CX3625CX.

## Author contributions

Y.D. acquired the funding and conceptualized the study. B.L., W.Z., and Y.D. developed the methodology. B.L. and W.Z. developed the software. B.L. and Y.L. performed the experimental studies. B.L. and Y.L. conducted the formal data analyses. B.L. completed the data visualization. B.L. wrote the original draft. Y.D. reviewed and edited the manuscript.

## Competing interests

The authors declare no competing interests.
