## [Peer Review File · Nature Communications]

Bilateral human laryngeal motor cortex in perceptual decision of lexical tone and voicing of consonantREVIEWER COMMENTS

Reviewer #1 (Remarks to the Author):

Key results

This study used the transcranial magnetic stimulation (TMS) technique applied to bilateral human laryngeal motor cortices (LMC) and the two-alternative forced choice (2-AFC) discrimination method to investigate whether the LMC (especially the dorsal LMC) is engaged in Mandarin speech perception as in production. Experiment 1 is more like an exploratory experiment that validates the methodology, and provided primary results that support the hypotheses. Experiment 2 is an improvement of Experiment 1 and also replicated its main results. Using psychometric fitting methods, the authors showed that TMS applied to the LMC affected the perception of both Mandarin lexical tone and constant. Moreover, the authors applied drift-diffusion models (DDM) in Experiment 2 and found that the LMC is engaged in all stages of the perceptual decision. Furthermore, the authors combined the slope and the DDM model results and found that the engagement of the LMC in Mandarin speech perception is left-biased and also modulated by task demands such that mid-level difficulty optimizes the involvement.

This study provides new knowledge to the field of speech perception and perceptual decision. The manuscript is clearly and concisely written and the visualization is good. However, I still have some comments that may help the authors to improve the quality of the manuscript.

Validity

This study includes two experiments with different samples and showed similar results. Moreover, Experiments 1 and 2 applied different stimulation protocols and multiple baseline controls that avoided methodological misinterpretations. Furthermore, the authors applied different behavioral analysis pipelines to improve the validity of data analyses.

There are indeed shortcomings of the study in terms of validity. The sample size of trials in Experiment 1 after separating the trials into three categories is small. This may be the reason why the authors got inconsistent results across conditions in Experiment 1. However, Experiment 2 replicated the main findings of Experiment 1 with an acceptable trial size. Another shortcoming is that the preprocessing procedure and outlier replacement strategy used in psychometric curve fitting is uncommon, but the authors also provided the results without replacement in the supplementary data, and the main findings were not changed.

Significance

This study provides new evidence to show that the laryngeal motor cortex is engaged in speech perception as in speech production. Three key findings are new to the speech perception field. First, the dorsal LMC is engaged in voice and perception as it is in production. Second, although the left LMC is more essential in speech perception corresponding to the popular models, its right counterpart is also important, especially in perceptual decision-making. Third, the left and right LMC is dynamically engaged in speech perception and different stages of decision-making as a function of task difficulty.

This study is a good extension of the existing theories and models (Hickok et al., 2011, *Neuron*, 69; Hickok & Poeppel, 2007, *Nature Reviews Neuroscience*, 8; Pulvermüller & Fadiga, 2010, *Nature Reviews Neuroscience*, 11.), especially in terms of the somatotopic involvement of the LMC, the engagement of the bilateral motor cortex, how the engagement is changed by cognitive demands, and what latent cognitive processes of perceptual decision the motor cortex is engaged in.

Data and methodology

Here are some questions related to the data and methodology that others can state more clearly.

1. Why consonant perception was more affected by cTBS in Exp 2 compared with tone perception? This is shown in Fig.3 where constant perception seemed to be more extensively affected by simulation. Also, in Fig.4, response bias was affected by cTBS only in consonant perception in noise task but not tone perception.

2. Is it possible that the LMC is also involved in the perception of the features that have not been tested in the current study (e.g., bilabial consonants)?

3. If the LMC encodes pitch, what kind of pitch features does it encode? The absolute pitch values or the pitch contours? The authors stated that it is the pitch contours that the LMC retrieves, but is there a possibility for the alternative?

4. In the hypothetical results stated in Fig.1d right, iTBS is also hypothesized to exert excitatory effects. However, there seem not to be any positive results from iTBS and no comments were written.

Analytical approach

1. Why did the authors use a logistic model with four parameters and constrained the upper and lower boundaries with a preceding free fitting? What is the advantage of using such a fitting method compared with the more commonly used method that fixes the boundaries as zero and one, or why is it necessary?

2. Why did the authors choose DDM instead of the other types of sequential sampling models? For example, the linear ballistic accumulator model.

3. Linear mixed models were used to interpret the TBS effects on reaction times. The models assumed that TBS had fixed effects on the slope, but is it possible that it also had random effects on the slope? That is, would TBS affect different subjects to different extents?

Clarity and context

This manuscript is clearly written, but I have some suggestions that may help with further improvement.

1. Page 16, lines 379–381 in Discussion. “Here we provide causal evidence for shared representations of the dLMC in the perceptual decision of phonetic features with its roles in pitch regulation and voicing control during articulation.” The term “representation” seems unsuitable. If the motor cortex in speech perception works as in production, it may provide efference copy instead of representing an object.

2. Page 16, lines 388–391 in Discussion. “For DDM parameters that represent latent psychological processes in decision-making, cTBS widened thresholds for decision boundary (a) across all conditions, and increased drift rates (v) for conditions with the left dLMC stimulated”. According to the results (Fig.

4), cTBS changed a in most conditions, but not all; cTBS changed but not increased all the drift rates for the left stimulation conditions. Also, it is better to have some comments on reaction times.

3. There are many acoustic and linguistic terms in the manuscript that may be difficult for most readers (e.g., lexical tone, lexical pitch, nonlinguistic, paralinguistic, and tone). Some may be different essentially while others may refer to the same term appearing in different contexts. In some situations, this makes it difficult to interpret the exact contributions of the current study compared with the previous ones. For example, on page 17, lines 400–402, “More recently, ‘virtual lesion’ studies by TMS have reported that the dLMC is causally involved in the discrimination of non-linguistic vocal pitch, singing voice, and paralinguistic prosody”. Some explanations of the differences between the terms may help clarify the expression.

4. Page 20, lines 489–505. “Our study provides a more sophisticated perspective ...” Although the authors tried their best to organize and interpret their results, it is still very cognitively demanding for me to get the whole picture (my motor cortex may also be engaged). A graph that shows the main findings of the current study is suggested. Although the authors have already provided a schematic illustration in Fig. 1a, it does not include the hemispheric asymmetry and modulations from task difficulty.

5. The authors could also have some comments on the prevailing models in sensorimotor integration of speech perception. For example, how the current results agree or disagree with the models, or what new knowledge is added to the models.

Reviewer #2 (Remarks to the Author):

The manuscript describes an interesting study of the role played by the human laryngeal motor cortex in speech perception. In more detail, the authors explore perceptual decisions in the context of lexical intonation and voicing. To this end, the authors conducted two TMS studies (one with an online protocol and one offline + fMRI localizer) to demonstrate a predominantly left lateral effect on RT and, through a drift diffusion model, also the impact of brain stimulation on decision-making subprocesses.

The authors have collected a significant amount of data, in well-designed studies, followed by a series of well-designed analyses. However, I have two main concerns. The first is related to the novelty of their results. The second relates to the specificity of the results. I will now argue in more detail.

Regarding the first point, the authors correctly cite all related literature in the discussion. However, the introduction reads as if no one had previously tested whether TMS to the motor system in general or specifically to the area of the larynx has an impact on vocal pitch discrimination tasks. As it stands now, the introduction constructs a narrative about novelty that, in my opinion, is not justified. Instead, the discussion does a better job referencing very similar literature (D'Ausilio et al., 2011; Leveque et al.,

2013; Sammler et al., 2015; Tang et al., 2021). However, my impression is that most of the authors' central findings had already been demonstrated by these previous studies. Left lateralization was demonstrated by Tang. The causal contribution of the larynx motor area on nonlinguistic vocal intonation, sung voice, and prosody was demonstrated by D'Ausilio, Leveque, and Sammler, respectively. Novelty relies on the use of lexical discrimination of intonation and voice. However, as I will argue in the following point, the reliability of these tasks also leaves some doubts.

The second point refers to the soundness of the results. Looking at the first study, I realize that the choice of stimuli followed a clear logic that unfortunately proved problematic for data analysis. As the authors suggest, the ambiguity of the unattended dimension made the interpretations less reliable.

However, if we look at the top row (a) of Figure 2, we can see the performance when the other dimension is unambiguous and thus, I suppose, somewhat more reliable. In this case, both stimulation sites on the left side (larynx and tongue M1) induce the same effect in the Consonant task (quiet stimuli), but not in the Tone task. This finding is problematic for three reasons. First, because it shows no site specificity for VOT processing, and second, tone processing is not affected by any of the stimulation sites. Despite these results, the second experiment uses stimulation of only the larynx motor cortex. Third, there appears to be no effect when noise is added, which has widely been shown to promote motor activities during speech perception and later, in experiment 2, is shown to induce the largest TMS-induced effects. The author instead, draw their conclusions based on extrapolations made on ambiguous and semi-ambiguous stimuli, which I do not consider a safe choice.

Then, in the second experiment, cTBS in the left laryngeal area induced an effect on the Consonant task with and without noise, while for the Tone task only with noise. Stimulation of the right side induced a reduction in performance only for the Consonant task when noise was added. This pattern, beside contradicting that of exp 1 (i.e., on the role of noise), to my mind, suggests a potential (uncontrolled) interaction with task complexity. In this regard, I am not sure that a constant level of noise has the same impact on both tasks and it would be important to know more.

Reviewer #3 (Remarks to the Author):

The manuscript reports a TMS experiment in which either the putative tongue motor cortex or putative larynx motor cortex were virtually lesioned and participants performed perceptual tests related consonant or lexical pitch perception. The methodology is of exceptionally high quality and is reported explicitly and clearly. The authors conclude that virtual lesions to these motor areas impair relevant perception. However, I have several concerns about interpretation which are detailed below. Most notably, the brain coordinates at which TMS was performed correspond poorly with the known locations of the brain areas that the authors purport to have studied. This presents a considerable obstacle as it is unclear at this time whether the authors have studied the same brain regions as the

literature against which they have compared their findings. This is an otherwise impressive and soundly conducted experiment that may be publishable with major revisions.

Major comments

Throughout - Is it insufficiently clear whether the manuscript concerns primary motor cortex (central sulcus, BA 4) or premotor cortex (precentral gyrus, BA 6). It is a habit of the TMS/perception literature to remain ambiguous on this point, but the distinction is an important one that I would urge the authors to be explicit about throughout the manuscript.

L 662 -TMS motor thresholds were estimated using standard and well accepted protocols. However, it seems that thresholds were calculated on a day separate from experimental testing. These thresholds are known to vary considerably between sessions as they are sensitive to small differences in localisation (how closely hand M1 was targeted), coil orientation (typically peaking at 45 degrees from the midline), as well as participant biological factors (sleep, caffeine, etc). Please elaborate on to what extent these thresholds should be taken as valid indicators of sensitivity to stimulation across test days and to what extent this should influence the interpretation of the results.

L 611 – “48 participants performed larynx-related (say [a]) and tongue related (say [th 612]) articulation tasks (dLMC: [± 40 , -5, 50], TMC: [± 59 , -3, 36]”. I commend the authors on pre-localising targets for your test population, however these coordinates are very surprising. Based on the prior literature I would have expected to find the dLMC at approximately either [-42, -12 38] or [-50 , -6, 46] for the primary motor cortex (deep in the central sulcus, BA 4) or pre-motor cortex (gyral surface, BA 6), respectively. Likewise, I might have expected tongue motor cortex to be quite a lot more ventral at about [54, -8, 28]. Given the coordinates reported in the paper, I might have expected this to be more akin to M1-lip.

Admittedly, those estimates are from samples that skew towards Caucasian and there are known differences between Caucasian as compared to Asian template brains, it seems unlikely that differences in template alone would cause such large disparities in localisation. While clearly the localisation study indicates that this coordinate is relevant for vowel production, it is not clear that dLMC is a credible interpretation.

The interpretation in the discussion section is difficult to evaluate without greater certainty around localisation.

L424 – This paragraph makes several claims that are vague, over-reaching and/or not supportable by the references provided. Please revise.

Discussion – The focus here on the dorsal stream (usually conceived of as STG->IFG via the arcuate fasciculus) and the ventral stream (usually conceived of as the ITG/MTG -> IFG via the uncinata fasciculus) does not seem well motivated. The manipulations in the present study are at (pre?)-motor cortex and well downstream of this distinction. I do not see how this is a particular issue on which the present study provides new information.

Minor comments

Figure 1a. The “ “ around “mirrors” leads one to suspect that this is an allusion to mirror neurons. As this concept has been habitually overstated and under-evidenced I would suggest omitting it here unless the authors wish to make an explicit case that they are studying mirror neurons. The figure itself is excellent.

L466 knowledge

L 818 Fourth.

We thank the editor and three reviewers for giving constructive advices. To account for the critiques and suggestions, we have substantially revised the manuscript. The following text shows our point-by-point responses to your concerns and questions. With your help, we believe that the manuscript has been improved and can better demonstrate our main discoveries, and we hope that all the concerns have been addressed satisfactorily. In the revised manuscript, all changes are marked in red. Thank you for the opportunity to resubmit our work.

Reviewer #1:

Key results

This study used the transcranial magnetic stimulation (TMS) technique applied to bilateral human laryngeal motor cortices (LMC) and the two-alternative forced choice (2-AFC) discrimination method to investigate whether the LMC (especially the dorsal LMC) is engaged in Mandarin speech perception as in production. Experiment 1 is more like an exploratory experiment that validates the methodology, and provided primary results that support the hypotheses. Experiment 2 is an improvement of Experiment 1 and also replicated its main results. Using psychometric fitting methods, the authors showed that TMS applied to the LMC affected the perception of both Mandarin lexical tone and constant. Moreover, the authors applied drift-diffusion models (DDM) in Experiment 2 and found that the LMC is engaged in all stages of the perceptual decision. Furthermore, the authors combined the slope and the DDM model results and found that the engagement of the LMC in Mandarin speech perception is left-biased and also modulated by task demands such that mid-level difficulty optimizes the involvement.

This study provides new knowledge to the field of speech perception and perceptual decision. The manuscript is clearly and concisely written and the visualization is good. However, I still have some comments that may help the authors to improve the quality of the manuscript.

Validity

This study includes two experiments with different samples and showed similar results. Moreover, Experiments 1 and 2 applied different stimulation protocols and multiple baseline controls that avoided methodological misinterpretations. Furthermore, the authors applied different behavioral analysis pipelines to improve the validity of data analyses.

There are indeed shortcomings of the study in terms of validity. The sample size of trials in Experiment 1 after separating the trials into three categories is small. This may be the reason why the authors got inconsistent results across conditions in Experiment 1. However, Experiment 2 replicated the main findings of Experiment 1 with an acceptable trial size. Another shortcoming is that the preprocessing procedure and outlier replacement strategy used in psychometric curve fitting is uncommon, but the authors also provided the results without replacement in the supplementary data, and the main findings were not changed.

Response:

Thank you for your good summary and positive comments.

As you mentioned, the reason why we applied two experiments with similar protocols and two different behavioral analysis pipelines (for Experiment 2) is to increase the validity of the study. We now explicitly pointed out that Experiment 1 is exploratory (Line 110) and that “Since Experiment 1 was exploratory, we interpret our results in the following discussion based solely on the findings from Experiment 2.” (Line 323-325)

The psychometric curve fitting procedure in the current study is indeed not common. Because we simultaneously applied background noise and acoustic continuum to increase the task difficulty, when TMS was additionally applied, participants' abilities to identify phonemes would be severely hampered and thus invalid slopes from illy-fitted psychometric curves would be derived. Instead of deleting these slopes, we chose to replace them with floor values as they were still meaningful in representing participants' deteriorated abilities due to brain stimulation. Results before such replacement procedure were shown in Supplementary Fig. 2, and as you mentioned, the main findings were not changed.

Significance

This study provides new evidence to show that the laryngeal motor cortex is engaged in speech perception as in speech production. Three key findings are new to the speech perception field. First, the dorsal LMC is engaged in voice and perception as it is in production. Second, although the left LMC is more essential in speech perception corresponding to the popular models, its right counterpart is also important, especially in perceptual decision-making. Third, the left and right LMC is dynamically engaged in speech perception and different stages of decision-making as a function of task difficulty.

This study is a good extension of the existing theories and models (Hickok et al., 2011, *Neuron*, 69; Hickok & Poeppel, 2007, *Nature Reviews Neuroscience*, 8; Pulvermüller & Fadiga, 2010, *Nature Reviews Neuroscience*, 11.), especially in terms of the somatotopic involvement of the LMC, the engagement of the bilateral motor cortex, how the engagement is changed by cognitive demands, and what latent cognitive processes of perceptual decision the motor cortex is engaged in.

Response:

Thank you for your recognition of the theoretical implications of the current study. We revised the manuscript to better demonstrate potential contributions to the current theories, which we proposed in both spatial and temporal domains.

In the spatial domain, we proposed four hypotheses (i.e., the acoustic, lexical, motor, and redundancy hypotheses) on the functions of bilateral motor cortices in speech perception. We illustrate the four hypotheses in new **Fig. 1b-e** and explain them

in details in the Introduction (Line 67-88): “The acoustic hypothesis posits that bilateral motor cortices process speech in ways resembling the auditory cortices, with the left motor cortex being more sensitive to temporal modulation and the right counterpart more attuned to spectral modulation^{12,13}. If so, the perception of consonant with temporally fast-varying VOT is expected to be left-lateralized, whereas that of lexical tone with spectrally fast-changing contour is expected to be right-dominant (Fig. 1b). In contrast, the lexical hypothesis suggests that the left motor cortex is more involved in lexical processing, and hence left lateralization is expected for both lexical tone and consonant perception (Fig. 1c). The motor hypothesis suggests that the motor cortices generate an "internal model" in speech perception as if they are enrolled in articulation with bilateral effector specificity^{14,15}. Since lexical tones are determined by laryngeal movements (pitch regulations) and articulating dental plosives recruits both the larynx (voicing on/offset) and tongue (dental consonants)¹⁶, based on the motor hypothesis, lexical tone perception would engage the dLMC, whereas consonant perception would enroll both the dLMC and TMC (Fig. 1d). Finally, the redundancy hypothesis proposes that the left motor cortex is more fundamental in speech perception, with its right counterpart being redundant and optimized only when the left is insufficient for completing perception (Fig. 1e). Lesions¹⁷ or “virtual lesions”^{18,19} studies have shown that perturbation of the left language areas triggers compensatory activations in the right counterparts. Background noise was added to half of the conditions and acoustic continuum was used to estimate the redundancy hypothesis by comparing the engagement of bilateral motor cortices under varying task difficulties”.

In the temporal domain, we proposed in the Introduction (Line 89-95): “For the third (temporal) question, speech perceptual decision is postulated as a three-stage procedure encompassing extraction of acoustic-phonetic features, mapping to phonemic categories, and response selection, in which the (left) motor cortex is interactively involved in all stages along with the auditory cortices²⁰. Although less is known about the right motor cortex, we hypothesized that it may be involved in all temporal stages in speech perceptual decision due to the functional symmetry to the left counterpart and its sub-regions during articulation^{14,15}.”.

We summarize our findings and corresponding theoretic contributions in the Conclusion (Line 456-467) and illustrate the model in new **Fig. 5**: “In conclusion, as illustrated in Fig. 5, we propose a model of bilateral dLMC engagement in the temporal dynamics of perceptual decision for lexical speech cues. Firstly, the recruitment of

bilateral dLMC for the perceptual decision of Mandarin lexical tone and voicing of consonant confirms the motor hypothesis regarding bilateral effector-specificity. Secondly, the left-dominance of the dLMC and the compensatory engagement of its right counterpart under increased difficulty support the redundancy hypothesis. Thirdly, bilateral dLMC are involved in multiple stages of the perceptual decision-making process, which is mediated by the hemisphere and cognitive demands. This study provides strong support for the sensorimotor integration account of speech perception^{2,3}. Moreover, our findings broaden the scope of research on this topic by providing insight into the functional spatial distributions of bilateral motor cortices and their involvement in the temporal dynamics of speech perceptual decision-making”.

Fig. 5. The proposed model of bilateral dLMC engagement in the perceptual decision of speech cues.

Bilateral human dLMC are engaged in multiple stages of the perceptual decision process of Mandarin lexical tone and voicing of consonant as a function of task difficulty. In the spatial domain, this supports the motor hypothesis (with bilateral effector-specific engagement) and the redundancy hypothesis (with left

dominance and right upregulation as difficulty increases). In the temporal domain, this reveals the recruitment of the dLMC in multiple steps of perceptual decision, which is also modulated by the hemisphere and task demands.

(a–c) The involvement of the left dLMC in quiet (a), in noise (b), and when the speech stimulus is completely ambiguous (c).

(d) Perceptual decision process depicted by hierarchical drift-diffusion model (HDDM) with three determining parameters: boundary threshold a , drift rate ν , and starting point or response bias z .

(e–g) The involvement of the right dLMC in quiet (a), in noise (b), and when the speech stimulus is completely ambiguous (g).

Semicircles in (a–c) and (e–g) demonstrate evidence for the effector-specific engagement (whether or not engaged in tone and/or consonant perception) of the left/right dLMC at each difficulty level, as revealed by cTBS modulation. Opaque semicircles: cTBS changed the psychometric slope (perceptual sensitivity); semitransparent semicircles: cTBS only changed the RTs or DDM parameters; transparent semicircles: no cTBS effects were found. orange: consonant tasks; blue: lexical tone tasks.

Spotlights from (a–c) and (e–g) to (d): engagement of the dLMC in affecting the stages (parameters) of the perceptual decision at different difficulty levels. Purple: boundary threshold a ; blue: drift rate ν ; orange: response bias z .

This illustration is based only on cTBS results from Experiment 2. See Table 1 for a quantitative summary.

Data and methodology

Here are some questions related to the data and methodology that others can state more clearly.

1. Why consonant perception was more affected by cTBS in Exp 2 compared with tone perception? This is shown in Fig.3 where constant perception seemed to be more extensively affected by simulation. Also, in Fig.4, response bias was affected by cTBS only in consonant perception in noise task but not tone perception.

Response:

We indeed observed stronger cTBS effects on consonant tasks compared with lexical tone tasks, although we balanced the difficulty between the two tasks by estimating and determining the individualized signal-to-noise ratio separately for consonants and tones. One possible reason is that plosive consonants featured by aspiration noise are acoustically much more indistinguishable from background noise than the pitch contour in lexical tone, and our balancing method may not be sufficient to counteract such a difference. Nevertheless, this asymmetry does not affect our main conclusions that the dLMC is causally and somatotopically engaged in speech perception. We added the statement to the results of Experiment 2 (Line 190–196):

“Note that cTBS had a more extensive impact on consonant perception than on lexical tone perception. Despite our efforts to balance the individualized SNR levels for both tasks (see Methods, Stimuli presentation), plosive consonants (featured by the length

of the aspiration noise resembling the masking noise) are less resilient to noise and more challenging to identify in noise (most participants performed unsatisfactorily in the consonant-in-noise perception task; see Supplementary Fig. 2)”.

2. Is it possible that the LMC is also involved in the perception of the features that have not been tested in the current study (e.g., bilabial consonants)?

Response:

According to Cheung et al. (2016), the motor cortex may represent the acoustic features of speech, regardless of the articulatory effectors. However, our results clearly showed that the dLMC is somatotopically involved in speech perception resembling its role in articulation. Moreover, according to previous functional localization studies (Brown et al., 2008; Eichert et al., 2020), the lip motor area is close to the dLMC in the current study that TMS upon the dLMC would unavoidably affect the lip area. Thus, to control the confounding effects, we intentionally avoided using lip sounds such as bilabial consonants (e.g., [b] or [p]). We added the statement to the results of Experiment 1 (Line 130–132):

“Meanwhile, to avoid confounding the examination of the motor somatotopy, labial consonants were intentionally excluded, as they may activate the lip motor area, which is in close proximity to our targeted regions¹⁴”.

References:

- Brown, S., Ngan, E., & Liotti, M. (2008). A larynx area in the human motor cortex. *Cerebral Cortex*, 18, 837–845. <https://doi.org/10.1093/cercor/bhm131>
- Cheung, C., Hamilton, L. S., Johnson, K., & Chang, E. F. (2016). The auditory representation of speech sounds in human motor cortex. *ELife*, 5, e12577. <https://doi.org/10.7554/eLife.12577>
- Eichert, N., Papp, D., Mars, R. B., & Watkins, K. E. (2020). Mapping human laryngeal motor cortex during vocalization. *Cerebral Cortex*, 30, 6254–6269. <https://doi.org/10.1093/cercor/bhaa182>

3. If the LMC encodes pitch, what kind of pitch features does it encode? The absolute pitch values or the pitch contours? The authors stated that it is the pitch contours that the LMC retrieves, but is there a possibility for the alternative?

Response:

Thank you for pointing out this possibility. Indeed, the lexical tone pair (tone1 and tone2) we used not only differed in pitch contour but also absolute pitch (see Supplementary Table 1). Note that, to keep the naturalness of the stimuli (Mandarin tone1

and tone2 are matched at the ending point), we did not balance the absolute pitch. Therefore, the current study could not exclude the possibility that the dLMC encodes pitch height in addition to the pitch contour. Nevertheless, it is the pitch contour that differentiates Mandarin lexical tones (Chao, 1980), and hence we followed this assumption in our manuscript. Although it does not affect our conclusion that the dLMC is engaged in distinguishing lexical pitch features, we added this limitation in the Discussion (Line 436–440):

“Secondly, while we demonstrated that the dLMC was recruited in distinguishing lexical pitch contours (level vs. rising tone), it is also possible that the dLMC encodes merely the mean F0 value that also differentiates the two tones (see Supplementary Table 1). Although this does not affect our main conclusions, it is an interesting topic for future studies to dissociate the dLMC functions for pitch height and pitch contour perception”.

References:

Chao, Y. R. (1980). A system of “tone-letters.” *Fangyan[Dialect]*, 2, 81–83.

4. In the hypothetical results stated in Fig. 1d right, iTBS is also hypothesized to exert excitatory effects. However, there seem not to be any positive results from iTBS and no comments were written.

Response:

Indeed, we did not find stable iTBS effects in both psychometric curve fitting and HDDM parameters. One possibility is that the contributions of the motor cortex were optimized in difficult tasks and thus could not be further increased by brain stimulation, but we have confirmed our hypotheses using the “virtual lesion mode” by cTBS. We added this limitation in the Discussion (Line 441–447):

“As for the TMS methodology, we only found the “virtual lesion” effects by cTBS but not any excitatory iTBS effects. It is possible that the motor cortices were already fully recruited and reluctant to further enhancement. Indeed, most previous TMS studies have only reported inhibitory effects on the motor engagement in speech perception (e.g.,^{10,50}), suggesting that speech perception improvement through TMS on the motor cortex may be challenging. Further studies exploring cognitive enhancement through TMS are necessary for clinical applications.”.

Analytical approach

1. Why did the authors use a logistic model with four parameters and constrained the

upper and lower boundaries with a preceding free fitting? What is the advantage of using such a fitting method compared with the more commonly used method that fixes the boundaries as zero and one, or why is it necessary?

Response:

A standard psychometric function is a four-parameter function that contains an upper asymptote, a lower asymptote, a point of Subjective Equity (PSE), and the slope of the curve at the PSE (below shows the section from Kingdom & Prins, 2016, p. 79).

4.3.2 Details of Function Types

Above we derived the generic formulation of the PF (Section 4.3.1.1):

$$\psi(x; \alpha, \beta, \gamma, \lambda) = \gamma + (1 - \gamma - \lambda)F(x; \alpha, \beta) \quad (4.2b)$$

As discussed there, under the high-threshold detection model, $F(x; \alpha, \beta)$ describes the probability of detection of the stimulus by the underlying sensory mechanism as a function of stimulus intensity x , γ corresponds to the guess rate (the probability of a correct response when the stimulus is not detected by the underlying sensory mechanism), and λ corresponds to the lapse rate (the probability of an incorrect response, which is independent of stimulus intensity).

Several functions are in use for $F(x; \alpha, \beta)$. We list here the most commonly used functions. Examples of all of these are shown in Figure 4.3. We will consistently use the symbol α to denote the location parameter (threshold) and the symbol β to denote the rate-of-change or slope parameter, even where this flies in the face of convention. We will also use expressions of F in which increasing values of β correspond to increasing slopes of F , even if this defies convention. Box 4.1 explains how the routines that implement the PFs in the Palamedes toolbox are used.

Indeed, most studies in this field used a simplified version of the psychometric function that directly fixed the upper and lower asymptotes to 1 and 0 (logistic function) (e.g., Xu et al., 2006). This is reasonable if they assumed that participants performed with high accuracy when the acoustic signal was at the clear ends of a continuum, which is usually true if no extra acoustic or cognitive disturbance exist as phoneme identification itself is not a cognitively demanding task. However, in our current study, for half of the trials, background noise masked the acoustic signal even when it was phonetically unambiguous, and thus it is unreasonable to still assume that participants' performance could reach the ceiling. We therefore set the upper and lower asymptotes as fitting parameters to avoid this underfitting. Meanwhile, to prevent over-fitting, we fixed the two asymptotes at group level for each condition after free fitting (i.e., following the procedure in Sammler et al., 2015). We added the statement to the fitting psychometric function in the Methods (Line 677–680).

“We used a four-parameter function⁷¹ instead of a more widely used simplified version that fixes the lower and upper asymptotes to 0 and 1. This is because it is unreasonable to assume that participants could perform 100% correctly even when the tone or consonant is unambiguous, especially in the noise-masking conditions”.

References:

- Kingdom, F. A. A., & Prins, N. (2016). *Psychophysics: A Practical Introduction* (M. Tucker (ed.)). Academic Press.
- Sammler, D., Grosbras, M.-H. H., Anwander, A., Bestelmeyer, P. E. G., & Belin, P. (2015). Dorsal and ventral pathways for prosody. *Current Biology*, 25(23), 1–7. <https://doi.org/10.1016/j.cub.2015.10.009>
- Xu, Y., Gandour, J. T., & Francis, A. L. (2006). Effects of language experience and stimulus complexity on the categorical perception of pitch direction. *Journal of Acoustical Society of America*, 120(2), 1063–1074. <https://doi.org/10.1121/1.2213572>

2. Why did the authors choose DDM instead of the other types of sequential sampling models? For example, the linear ballistic accumulator model.

Response:

We used DDM rather than LBA because DDM assumes only one stochastic accumulator that better fits our experimental paradigm, compared with the LBA which assumes two ballistic accumulators. We added the statement to the HDDM and reaction time analysis in the Methods (Line 708–720):

“Single-trial data (binary decision and reaction times) were entered into HDDM model²² using the HDDM package²¹ based on Python 2.7 to disentangle TBS effects on three key parameters of perceptual decision-making: starting point or response bias (z), average drift rate (v) of evidence accumulation, and threshold between decision boundaries (a). We used HDDM, instead the equivalently prevailing linear ballistic accumulator (LBA) model that also belongs to the sequential sampling family, for two reasons. First, HDDM assumes only one accumulator that samples evidence and thus better fits our identification paradigm where the two alternatives varied only in one feature (pitch contour or VOT), compared with the LBA with two accumulators. Second, HDDM can be regarded as the signal detection theory with temporal dynamics as it assumes a stochastic process for evidence accumulation. This matches the goals of our psychophysical experiment to detect subjects' ability to segregate acoustic information from ambiguous signal and noise.”

References:

- Mulder, M. J., van Maanen, L., & Forstmann, B. U. (2014). Perceptual decision neurosciences - a model-based review. *Neuroscience*, 277, 872–884. <https://doi.org/10.1016/j.neuroscience.2014.07.031>

3. Linear mixed models were used to interpret the TBS effects on reaction times. The models assumed that TBS had fixed effects on the slope, but is it possible that it also had random effects on the slope? That is, would TBS affect different subjects to different extents?

Response:

Indeed, we used linear mixed models to test whether TBS upon the dLMC exerted significant effects on reaction times regardless of whether the effects are equivalent across subjects (fixed effects on the slope) or also contain variance (random effects on the slope). We, therefore, simplified the model to the extent that it can account for our hypothesis but also avoid over-fitting. Also, a model with random effects may induce fitting failure for convergence problems. We added the statement in Line 757–759 in the Methods: “Since our goal was to testify whether cTBS effects on slopes existed and to avoid failure of convergence, we did not model random effects on slopes”. Nevertheless, future studies are needed to explore TMS effects on reaction times in more detail by testing different linear mixed models.

Clarity and context

This manuscript is clearly written, but I have some suggestions that may help with further improvement.

1. Page 16, lines 379–381 in Discussion. “Here we provide causal evidence for shared representations of the dLMC in the perceptual decision of phonetic features with its roles in pitch regulation and voicing control during articulation.” The term “representation” seems unsuitable. If the motor cortex in speech perception works as in production, it may provide efference copy instead of representing an object.

Response:

Thank you. We reorganized our discussion and deleted this sentence in the revised version. We did mention the efference copy account in Line 416-418: “One possible interpretation of our findings is that bilateral dLMC provide an articulatory efference copy⁴⁸ that serves as categorical templates to match with sensory input”.

2. Page 16, lines 388–391 in Discussion. “For DDM parameters that represent latent psychological processes in decision-making, cTBS widened thresholds for decision boundary (a) across all conditions, and increased drift rates (v) for conditions with the left dLMC stimulated”. According to the results (Fig. 4), cTBS changed a in most conditions, but not all; cTBS changed but not increased all the drift rates for the left stimulation conditions. Also, it is better to have some comments on reaction times.

Response:

Thank you for your rigor, we replaced “all” with “most” for describing cTBS effects

on the boundary threshold as well as reaction times. Meanwhile, we replaced the term “increased” with “affected” when discussing cTBS effects on the drift rate. See Line 401–404 in the Discussion: “The effects of dLMC stimulation on RTs and HDDM boundary threshold a were consistent across most conditions, regardless of the hemisphere and task difficulty. However, the drift rate ν was only affected by left dLMC stimulation, but not by right stimulation”.

A short comment on reaction times is given in Line 249–251 in the Results: “This corresponds to the general increase in thresholds a as the decision boundary is closely related to the strategy of speed-accuracy tradeoff²⁹. For RT analyses in iTBS conditions, see Supplementary Fig. 6”.

3. There are many acoustic and linguistic terms in the manuscript that may be difficult for most readers (e.g., lexical tone, lexical pitch, nonlinguistic, paralinguistic, and tone). Some may be different essentially while others may refer to the same term appearing in different contexts. In some situations, this makes it difficult to interpret the exact contributions of the current study compared with the previous ones. For example, on page 17, lines 400–402, “More recently, ‘virtual lesion’ studies by TMS have reported that the dLMC is causally involved in the discrimination of non-linguistic vocal pitch, singing voice, and paralinguistic prosody”. Some explanations of the differences between the terms may help clarify the expression.

Response:

In the revised manuscript, we changed all “lexical pitch” to “lexical tone” and avoided using the terms “nonlinguistic” and “paralinguistic”. Rather, we only used “lexical” and “non-lexical” to distinguish speech cues that determine meanings or not. For example, Line 42–46, “Transcranial magnetic stimulation (TMS) studies have identified a causal engagement of the left motor cortex in speech perception in an effector-specific manner, such as the lip motor subregion for bilabial consonants^{5,6}, and the tongue motor area for dental consonants^{7,8} and vowels⁹, while the right motor cortex has been linked to non-lexical prosodic cues¹⁰”; Line 333–338, “Previous TMS studies have reported that the dLMC is causally involved in the discrimination of non-lexical vocal pitch^{10,31} and singing voice³². A recent study demonstrated that lexical tone processing is suppressed after stimulating a lip region close to the dLMC in the left motor cortex³³. The current study takes a step further and establishes a causal link between the human dLMC (defined by BOLD activations in articulation tasks) and perceptual decision of lexical tone”.

4. Page 20, lines 489–505. “Our study provides a more sophisticated perspective ...” Although the authors tried their best to organize and interpret their results, it is still very cognitively demanding for me to get the whole picture (my motor cortex may also be engaged). A graph that shows the main findings of the current study is suggested. Although the authors have already provided a schematic illustration in Fig. 1a, it does not include the hemispheric asymmetry and modulations from task difficulty.

Response:

We updated **Fig. 1** so that it not only shows the concept of sensorimotor integration (**Fig. 1a**), but also the hypothetical functional distributions of bilateral motor cortices and modulations from the task difficulty (spatial hypotheses, **Fig. 1 b–e**), as well as the hypothetical temporal stages of speech perceptual decision that the motor cortex contributes to (temporal hypotheses, **Fig. 1 i–l**).

Fig. 1. Experimental design and hypothesized results.

- (a) Schematic illustration of sensorimotor integration in Mandarin speech perception. The dLMC commands the larynx to control the pitch of voice and onset/offset of phonation during speech articulation. The dLMC is assumed to engage in the perception of lexical tone and consonant (VOT contrast) in Mandarin listeners, in a way of motor simulation as in speech production.
- (b–e) Predicted spatial patterns of bilateral motor engagement by four different hypotheses: the acoustic hypothesis (b), the lexical hypothesis (c), the motor hypothesis (d), and the redundancy hypothesis (e). See the main text for explanations of each hypothesis.
- (f–h) Hypothetical results for psychometric curve fitting. (f) How TMS may affect curves of identification: compared with baseline (grey line), inhibitory stimulation (rTMS and cTBS) would “flatten” the curve and impair performance (light purple line), while excitatory stimulation (iTBS) would “steepen” the curve and improve performance (dark purple line). Oblique lines crossing the point of subjective equality (PSE) represent slopes of the curves. We used the slope estimation method to test the hemispheric asymmetry and effector-specificity of the motor engagement. (g, h) How slopes of tone and consonant curves may be altered by TMS in Experiment 1 (g) and Experiment 2 (h) if (+) or if not (-) the dLMC is engaged, compared with sham (red dashed lines). Red arrows: inhibitory effect; white arrows: excitatory effect. The boxplots in (g) and (h) are schematic representations of possible slope distributions.
- (i–l) Temporal stages of motor engagement were tested by the hierarchical drift-diffusion model (HDDM). (i) How HDDM predicts distributions of reaction times. (j–l) Hypothetical models with the parameter a (boundary threshold, j), ν (drift rate, k), and z (starting point, l) modulated by TMS, respectively.
- For the whole figure: blue and orange represent lexical tone and consonant, respectively; light purple, dark purple, white, and grey represent inhibitory TMS (rTMS and cTBS), excitatory TMS (iTBS), TMC stimulation, and sham, respectively. These color patterns are the same in Fig. 2 and Fig. 3.
- MC: motor cortex; AC: auditory cortex; dLMC: dorsal laryngeal motor cortex; TMC: tongue motor cortex.

Also, we added a new **Table 1** for quantitative summary of the results and a new **Fig. 5** as a model that summarizes how bilateral dLMC are engaged in the perceptual decision of Mandarin lexical tone and voicing of consonant as a function of task difficulty.

Table 1. Modulations of cTBS effects by hemisphere and task difficulty in Experiment 2.

cTBS target	Indices	In quiet (Fig. 3, 4)		In noise (Fig. 3, 4)		Completely ambiguous (Supplementary Table 2)	
		Tone	Cons.	Tone	Cons.	Tone	Cons.
Left dLMC	Slope	-	+	+	+	n.a.	n.a.
	RT	+	+	-	+	n.a.	n.a.
	a	-	+	+	+	-	-
	v	+	+	+	+	-	-
	z	-	-	-	+	+	-
Right dLMC	Slope	-	-	-	+	n.a.	n.a.
	RT	+	+	+	+	n.a.	n.a.
	a	+	+	+	+	n.a.	-
	v	-	-	-	-	n.a.	-
	z	-	-	-	+	n.a.	n.a.

Results are adopted from Fig. 3 (cTBS effects on slopes), Fig. 4 (cTBS effects on HDDM parameters), and Supplementary Table 2 (interactions between cTBS effects on HDDM parameters and stimulus ambiguity).

Completely ambiguous: trials where the tone or consonant was at the categorical boundary (i.e., step3) of the continuum and completely ambiguous.

+: significant cTBS modulation effect; -: no significant cTBS effect; n.a.: not analyzed. dLMC: dorsal laryngeal motor cortex; Cons.: consonant; RT: reaction time; *a*: boundary threshold; *v*: drift rate; *z*: starting point.

Fig. 5. The proposed model of bilateral dLMC engagement in the perceptual decision of speech cues.

Bilateral human dLMC are engaged in multiple stages of the perceptual decision process of Mandarin lexical tone and voicing of consonant as a function of task difficulty. In the spatial domain, this supports the motor hypothesis (with bilateral effector-specific engagement) and the redundancy hypothesis (with left dominance and right upregulation as difficulty increases). In the temporal domain, this reveals the recruitment of the dLMC in multiple steps of perceptual decision, which is also modulated by the hemisphere and task demands.

(a–c) The involvement of the left dLMC in quiet **(a)**, in noise **(b)**, and when the speech stimulus is completely ambiguous **(c)**.

(d) Perceptual decision process depicted by hierarchical drift-diffusion model (HDDM) with three determining parameters: boundary threshold a , drift rate v , and starting point or response bias z .

(e–g) The involvement of the right dLMC in quiet **(a)**, in noise **(b)**, and when the speech stimulus is completely ambiguous **(g)**.

Semicircles in **(a–c)** and **(e–g)** demonstrate evidence for the effector-specific engagement (whether or not engaged in tone and/or consonant perception) of the left/right dLMC at each difficulty level, as revealed by cTBS modulation. Opaque semicircles: cTBS changed the psychometric slope (perceptual sensitivity); semitransparent semicircles: cTBS only changed the RTs or DDM parameters;

transparent semicircles: no cTBS effects were found. orange: consonant tasks; blue: lexical tone tasks.

Spotlights from (a–c) and (e–g) to (d): engagement of the dLMC in affecting the stages (parameters) of the perceptual decision at different difficulty levels. Purple: boundary threshold a ; blue: drift rate ν ; orange: response bias z .

This illustration is based only on cTBS results from Experiment 2. See Table 1 for a quantitative summary.

5. The authors could also have some comments on the prevailing models in sensorimotor integration of speech perception. For example, how the current results agree or disagree with the models, or what new knowledge is added to the models.

Response:

We have thoroughly rewritten our manuscript to emphasize the prevailing models, i.e., dual-stream models (Hickok & Poeppel, 2007) and the effector-specificity approach of speech perception (D’Ausilio et al., 2009), unanswered questions, our hypotheses, our conclusions, and how our findings add to the prevailing models.

For the description of the models, we summarized the main points at the start of the Introduction (Line 38–46): “Speech perception has long been hypothesized to recruit motoric simulation by the speech motor system, as posited by the motor theory of speech perception¹. Recent neuroanatomical models of speech processing propose that the left motor cortex maps phonological analyses onto motor representations, which may compensate for degraded auditory processing in challenging listening conditions^{2–4}. Transcranial magnetic stimulation (TMS) studies have identified a causal engagement of the left motor cortex in speech perception in an effector-specific manner, such as the lip motor subregion for bilabial consonants^{5,6}, and the tongue motor area for dental consonants^{7,8} and vowels⁹, while the right motor cortex has been linked to non-lexical prosodic cues¹⁰”.

We then pointed out the unanswered questions of the prevailing models (Line 46–52): “However, three outstanding questions regarding the role of the bilateral motor cortices in speech perception remain unanswered: 1) whether the laryngeal motor cortex (LMC) is engaged in an effector-specific manner similar to the lip and tongue areas, 2) how the bilateral motor cortices cooperate during speech perception under varying difficulty, and 3) what specific stages of the perceptual decision-making process the bilateral motor cortices modulate”.

Next, we proposed the hypotheses based on the mentioned questions (Line 53–107), followed by the design of experiments.

“This study aims to address the first and second questions (spatial questions) by

exploring four hypothetical mechanisms that may drive the functional distributions of bilateral motor cortices in speech perception: the acoustic hypothesis, the lexical hypothesis, the motor hypothesis, and the redundancy hypothesis (Fig. 1b-e). To do so, we delivered repetitive TMS (rTMS, Experiment 1) or theta-burst stimulation (TBS, Experiment 2, including intermittent TBS, i.e., iTBS, and continuous TBS, i.e., cTBS) to the left or right motor cortex (in Experiment 1, LMC and tongue motor cortex, TMC; in Experiment 2, the LMC only) of Mandarin speakers to investigate if the identification of lexical tone (tone1 vs. tone2, featured by pitch contour) and dental plosive consonant ([t] vs. [t^h], featured by voice onset time, VOT) in quiet or in noisy background would be modulated accordingly (Fig. 1f-h). To localize the dorsal LMC (dLMC), which is closely related to speech production¹¹, and TMC, respectively, participants underwent a functional magnetic resonance (fMRI) pretest where they performed phonation and tongue movement tasks.

The acoustic hypothesis posits that bilateral motor cortices process speech in ways resembling the auditory cortices, with the left motor cortex being more sensitive to temporal modulation and the right counterpart more attuned to spectral modulation^{12,13}. If so, the perception of consonant with temporally fast-varying VOT is expected to be left-lateralized, whereas that of lexical tone with spectrally fast-changing contour is expected to be right-dominant (Fig. 1b). In contrast, the lexical hypothesis suggests that the left motor cortex is more involved in lexical processing, and hence left lateralization is expected for both lexical tone and consonant perception (Fig. 1c). The motor hypothesis suggests that the motor cortices generate an "internal model" in speech perception as if they are enrolled in articulation with bilateral effector specificity^{14,15}. Since lexical tones are determined by laryngeal movements (pitch regulations) and articulating dental plosives recruits both the larynx (voicing on/offset) and tongue (dental consonants)¹⁶, based on the motor hypothesis, lexical tone perception would engage the dLMC, whereas consonant perception would enroll both the dLMC and TMC (Fig. 1d). Finally, the redundancy hypothesis proposes that the left motor cortex is more fundamental in speech perception, with its right counterpart being redundant and optimized only when the left is insufficient for completing perception (Fig. 1e). Lesions¹⁷ or "virtual lesions"^{18,19} studies have shown that perturbation of the left language areas triggers compensatory activations in the right counterparts. Background noise was added to half of the conditions and acoustic continuum was used to estimate the redundancy hypothesis by comparing the engagement of bilateral motor cortices

under varying task difficulties.

For the third (temporal) question, speech perceptual decision is postulated as a three-stage procedure encompassing extraction of acoustic-phonetic features, mapping to phonemic categories, and response selection, in which the (left) motor cortex is interactively involved in all stages along with the auditory cortices²⁰. Although less is known about the right motor cortex, we hypothesized that it may be involved in all temporal stages in speech perceptual decision due to the functional symmetry to the left counterpart and its sub-regions during articulation^{14,15}. To test this, we applied the hierarchical Bayesian estimation of the drift-diffusion model (HDDM)^{21,22} to single-trial binary responses and reaction times (RTs) in Experiment 2 to disentangle what latent dynamic decision processes the dLMC is engaged in (Fig. 1i-l).

Our results reveal an effector-specific involvement of bilateral dLMCs in perceptual decision of both lexical tone and voicing of consonant, lending support to the motor hypothesis. Meanwhile, we provide evidence for the redundancy hypothesis, as the left dLMC plays a dominant role, while the right counterpart is only crucial in challenging tasks. In contrast, the lexical hypothesis is only weakly supported, whereas the acoustic hypothesis is not confirmed. Moreover, the specific perceptual decision stages that are modulated by the dLMC hinge on the hemisphere and task difficulty. Taken together, these findings expand our knowledge of the underlying mechanisms and temporal dynamics of bilateral motor engagement in speech perceptual decision.”

We concluded our results in relation with our hypotheses (Line 456–463): “In conclusion, as illustrated in Fig. 5, we propose a model of bilateral dLMC engagement in the temporal dynamics of perceptual decision for lexical speech cues. Firstly, the recruitment of bilateral dLMC for the perceptual decision of Mandarin lexical tone and voicing of consonant confirms the motor hypothesis regarding bilateral effector-specificity. Secondly, the left-dominance of the dLMC and the compensatory engagement of its right counterpart under increased difficulty support the redundancy hypothesis. Thirdly, bilateral dLMC are involved in multiple stages of the perceptual decision-making process, which is mediated by the hemisphere and cognitive demands.”

Finally, we summarized how our findings may add knowledge to the current models (Lines 464–467): “This study provides strong support for the sensorimotor integration account of speech perception^{2,3}. Moreover, our findings broaden the scope of research on this topic by providing insight into the functional spatial distributions of bilateral motor cortices and their involvement in the temporal dynamics of speech

perceptual decision-making.”

Reviewer #2:

The manuscript describes an interesting study of the role played by the human laryngeal motor cortex in speech perception. In more detail, the authors explore perceptual decisions in the context of lexical intonation and voicing. To this end, the authors conducted two TMS studies (one with an online protocol and one offline + fMRI localizer) to demonstrate a predominantly left lateral effect on RT and, through a drift diffusion model, also the impact of brain stimulation on decision-making subprocesses.

The authors have collected a significant amount of data, in well-designed studies, followed by a series of well-designed analyses. However, I have two main concerns. The first is related to the novelty of their results. The second relates to the specificity of the results. I will now argue in more detail.

Response:

Thank you for your positive comments. Before point-by-point responses to your comments, we would like to have a gross statement first.

Regarding the novelty of the study, we admit that the original version overemphasized the causal role of the dLMC in the perception of lexical tone and VOT contrasts (the effector-specificity). But our findings also provide previously unreported evidence for the functional asymmetry of bilateral dLMC (and more generally, the motor cortex) engagement as well as the temporal dynamics of the dLMC involvement in speech perceptual decision. We thus thoroughly reorganized the Introduction and the Discussion sections and adjusted the Results section to better demonstrate the three key findings and the theoretic contributions to the current models (see Figure 1 and 5 for details).

Meanwhile, thank you for checking the results in detail. Indeed, the results are inconsistent if we take together the TMS effects on the slopes in Experiment 1 (for trials with the unattended dimension unambiguous) and cTBS effects on the slopes in Experiment 2. However, we regarded Experiment 1 as an exploratory study and only interpreted the results from Experiment 2 (in particular, cTBS conditions). Meanwhile, when testing hypotheses, we not only considered cTBS effects on the slopes of psychometric curves but also those on the hierarchical drift-diffusion model parameters and reaction times.

Regarding the first point, the authors correctly cite all related literature in the discussion. However, the introduction reads as if no one had previously tested whether TMS to the motor system in general or specifically to the area of the larynx has an impact on vocal pitch discrimination tasks. As it stands now, the introduction constructs a narrative about novelty that, in my opinion, is not justified. Instead, the discussion does a better job referencing very similar literature (D'Ausilio et al., 2011; Leveque et al., 2013;

Sammler et al., 2015; Tang et al., 2021).

Response:

The statement “A lip motor subregion is engaged ..., the contribution of the laryngeal motor cortex (LMC) to speech perception has rarely been investigated” in the original manuscript is indeed misleading. We reorganized the research questions in the Introduction (Line 46–52): “However, three outstanding questions regarding the role of the bilateral motor cortices in speech perception remain unanswered: 1) whether the laryngeal motor cortex (LMC) is engaged in an effector-specific manner similar to the lip and tongue areas, 2) how the bilateral motor cortices cooperate during speech perception under varying difficulty, and 3) what specific stages of the perceptual decision-making process the bilateral motor cortices modulate”.

However, my impression is that most of the authors' central findings had already been demonstrated by these previous studies. Left lateralization was demonstrated by Tang. The causal contribution of the larynx motor area on nonlinguistic vocal intonation, sung voice, and prosody was demonstrated by D'Ausilio, Leveque, and Sammler, respectively. Novelty relies on the use of lexical discrimination of intonation and voice. However, as I will argue in the following point, the reliability of these tasks also leaves some doubts.

Response:

Compared with D'Ausilio et al. (2011), Leveque et al. (2013), and Sammler et al. (2015) that found the motor engagement in the perception of non-lexical vocal pitch and singing voice, the novelty of the current study lies in demonstrating the dLMC engagement in lexical tone and voicing of consonant. More importantly, we, for the first time, depicted the hemispheric asymmetry of the bilateral dLMC engagement, as well as its temporal dynamics in the perceptual decision-making as a function of the hemisphere and task difficulty.

Tang et al. (2021) found left lateralization of motor engagement for lexical tone perception, but what we discovered was more sophisticated as although we found left lateralization for the cTBS effects on the slope of lexical tone judgment, the effects on reaction times and the HDDM boundary threshold a were bilateral and modulated by task difficulty for other parameters. Presumably, what Tang et al. (2021) discovered by the oddball paradigm is the engagement of the left motor cortex in the *pre-attentive detection* of acoustic changes in lexical tone, whereas what we found, using an explicit identification paradigm, is bilateral motor engagement (with left asymmetry) in the

perceptual decision of lexical tone (as well as voicing). We discussed this point in response to our proposed lexical and redundancy hypotheses (**Fig. 1c, e**) in Line 375–397 in the Discussion: “In contrast, our results only provide weak support for the lexical hypothesis, which suggests that the functional asymmetry of the motor cortex is due to differences in linguistic functions between lexical and non-lexical speech cues. This hypothesis contrasts with the acoustic hypothesis, which proposes that differences in stimulus acoustic properties (i.e., temporal and spectral modulation) drive the asymmetry¹². Previous studies have shown that the motor engagement for lexical and segmental units is left-dominant^{2,33,43}, whereas that for the non-lexical and suprasegmental prosody is biased to the right hemisphere^{10,33}. Tang et al. (2021)³³ used the oddball paradigm and found that linguistic functions drove the asymmetric auditory-motor processing of lexical tone in tonal and non-tonal language speakers, and the pre-attentive detection of lexical tone changes was left lateralized in tonal language speakers. Using an explicit identification paradigm, our study supports the lexical hypothesis but not the acoustic hypothesis by showing that the motor engagement for both lexical tone and consonant perception were left-biased. However, our evidence was weak as the observed hemispheric asymmetry was more sophisticated and modulated by task difficulty. Thus, the right motor cortex may be redundant instead of functionally specialized for non-lexical speech cues. Furthermore, contested results exist in the literature, as some studies found left motor activations during lexical tone perception in both tonal and non-tonal language speakers⁴⁴, but some found bilateral motor engagement during lexical tone perception with enhanced right auditory-motor connectivity⁴⁵. Overall, our findings add to the complex understanding of the neural mechanisms underlying motor asymmetry and suggest that further research is needed to fully elucidate this phenomenon”.

The second point refers to the soundness of the results. Looking at the first study, I realize that the choice of stimuli followed a clear logic that unfortunately proved problematic for data analysis. As the authors suggest, the ambiguity of the unattended dimension made the interpretations less reliable. However, if we look at the top row (a) of Figure 2, we can see the performance when the other dimension is unambiguous and thus, I suppose, somewhat more reliable.

Response:

Thank you for pointing out the problems of Experiment 1. We regarded Experiment 1 as an exploratory study and did not include it in the final interpretations. Here, we

explain our understanding on the results of Experiment 1. Indeed, we found competition between the perception of lexical tone and that of consonant when they were simultaneously ambiguous (see **Supplementary Fig. 3**). However, we separated the trials into three groups according to the ambiguity of the unattended dimension to avoid interactions between the unattended ambiguity and the perceptual decision. From our own perspective, the reliability of the results was reduced due to a relatively small number of trials after grouping, but not due to the interference from the unattended dimension. We thus prefer to consider the slope results in Experiment 1 from all three categories with different unattended ambiguities as equally valid but at different levels of difficulty. That is, we tended to keep taking all three categories, instead of only the unambiguous category of trials into consideration.

In this case, both stimulation sites on the left side (larynx and tongue M1) induce the same effect in the Consonant task (quiet stimuli), but not in the Tone task. This finding is problematic for three reasons.

Response:

We hope the following point-by-point responses and the revision of the manuscript have addressed all of your concerns.

First, because it shows no site specificity for VOT processing, and second, tone processing is not affected by any of the stimulation sites.

Response:

The lack of site specificity for VOT processing matches our expectation for the motor somatotopy in speech perception, as we stated in the Introduction (Line 74–81): “The motor hypothesis suggests that the motor cortices generate an “internal model” in speech perception as if they are enrolled in articulation with bilateral effector specificity^{14,15}. Since lexical tones are determined by laryngeal movements (pitch regulations) and articulating dental plosives recruits both the larynx (voicing on/offset) and tongue (dental consonants)¹⁶, based on the motor hypothesis, lexical tone perception would engage the dLMC, whereas consonant perception would enroll both the dLMC and TMC (Fig. 1d)”. Meanwhile, previous studies have found that the perceptual decision of VOT recruits both the tongue (Schomers et al. 2015) and (ventral) laryngeal motor areas (Tamura et al. 2022), which is consistent with our results. We explained why we chose the TMC in Line 119-122: “The TMC stimulation was set as

a site control in answering whether the motor engagement is effector-specific (the first spatial question), and verifying the paradigm as the engagement of the TMC in dental consonant perception has been well recognized^{6,7,23}.

For the processing of lexical tone, we indeed did not find significant TMS effects when the unattended dimension (i.e., consonant VOT) was unambiguous (**Fig. 2 c–f**), but significant inhibitory effects were found when the VOT was both half-ambiguous (**Fig. 2j**) and ambiguous (**Fig. 2m**). Meanwhile, for conditions with noise masking, more frequently, TMS upon the dLMC tended to exert inhibitory effects (TMS - sham < 0, **Fig. 2 f, j, m, n**). However, not all of them reached statistical significance. Note that, here we took the results from three categories as equally valid, and regarded speech perceptual decision in trials with different unattended dimension ambiguity as the same cognitive process with various task demands. Nevertheless, since the sample sizes were small, we did not use the results in Experiment 1 in the final interpretations.

References:

- Schomers, M. R., Kirilina, E., Weigand, A., Bajbouj, M., & Pulvermüller, F. (2015). Causal influence of articulatory motor cortex on comprehending single spoken words: TMS evidence. *Cerebral Cortex*, 25, 3894–3902. <https://doi.org/10.1093/cercor/bhu274>
- Tamura, S., Hirose, N., Mitsudo, T., Hoaki, N., Nakamura, I., Onitsuka, T., & Hirano, Y. (2022). Multi-modal imaging of the auditory-larynx motor network for voicing perception. *NeuroImage*, 251, 118981. <https://doi.org/10.1016/j.neuroimage.2022.118981>

Despite these results, the second experiment uses stimulation of only the larynx motor cortex.

Response:

We used only the dLMC stimulation in Experiment 2 because the current study mainly concentrated on the effector-specificity and the functional asymmetry of the engagement of bilateral dLMC in speech perception as well as the temporal dynamics. The purpose of applying TMS on the TMC in Experiment 1 was to validate the stimulation protocol as explained above. As shown, TMC stimulation results matched our expectations, we thus did not further apply TMC stimulation in Experiment 2.

Third, there appears to be no effect when noise is added, which has widely been shown to promote motor activities during speech perception and later, in experiment 2, is shown to induce the largest TMS-induced effects.

Response:

For the reasons described above, we here took all the results in Experiment 1 into consideration. Although not all conditions reached statistical significance, compared with conditions in quiet (**Fig. 2 c, d, g, h, k, l**), most conditions with noise revealed inhibitory TMS effects of the dLMC on lexical tone perception (i.e., rTMS - Sham < 0, **Fig. 2 f, j, m, n**). Meanwhile, if we assume that the enhanced ambiguity of the unattended dimension increased task difficulty, the fact that significant rTMS effects on lexical tone perception were only shown when the unattended dimension was half-ambiguous (**Fig. 2j**) and ambiguous (**Fig. 2m**) matched the scenario that cognitive demands increase the motor engagement. Also, it aligned with Experiment 2 where cTBS significantly modulated the psychometric slope of lexical tone identification only when noise was added. However, TMS effects on consonant perception seemed to be less stable. The limited sample sizes may account for this variability, and therefore we tried to replicate the major results in Experiment 1 by conducting Experiment 2.

The author instead, draw their conclusions based on extrapolations made on ambiguous and semi-ambiguous stimuli, which I do not consider a safe choice.

Response:

We admit that it is unsafe to draw the major conclusions based on Experiment 1 results, regardless of the ambiguity of the unattended dimension. Thus, we only took Experiment 2 (particularly the cTBS effects) into interpretations and discussions. Nevertheless, we reported Experiment 1 as it was a tentative study and also validated our paradigms. We highlighted our perspectives on the usage of the two experiments in the beginning of the Discussion (Line 323–325): “Since Experiment 1 was exploratory, we interpret our results in the following discussion based solely on the findings from Experiment 2”.

Moreover, in terms of the conclusions of Experiment 1, we believe that it is better to take into account all groups of data regardless of the ambiguity of the unattended dimension. First, although the perception of lexical tone was found to interfere with that of consonant and vice versa, the two cues are acoustically independent. Thus, the interference may occur in high-level cognitive domains other than the auditory analyses the current study concentrated on. It is, therefore, safe to assume that the underlying perceptual processes are similar regardless of the ambiguity of the unattended dimension. On the other hand, the ambiguity of the unattended dimension may

modulate task difficulty as the perceptual uncertainty in both dimensions may require more cognitive resources, instead of changing the perception itself and reducing its reliability.

Then, in the second experiment, cTBS in the left laryngeal area induced an effect on the Consonant task with and without noise, while for the Tone task only with noise. Stimulation of the right side induced a reduction in performance only for the Consonant task when noise was added. This pattern, beside contradicting that of exp 1 (i.e., on the role of noise), to my mind, suggests a potential (uncontrolled) interaction with task complexity. In this regard, I am not sure that a constant level of noise has the same impact on both tasks and it would be important to know more.

Response:

We admit that both Experiment 1 and Experiment 2 found unbalanced TMS effects on consonant and lexical tone perception. Experiment 1 found that lexical tone perception tended to be inhibited by rTMS in noisy and difficult conditions but the effects on consonant perception had relatively inconsistent patterns (Fig. 2). Experiment 2 found that cTBS exerted stronger and more extensive effects on consonant tasks compared with tone tasks. First, differences between the unbalanced patterns in Experiment 1 and Experiment 2 may result from differences in strategies to balance task difficulty. In Experiment 1, a fixed signal-to-noise ratio (SNR) level for noise masking was applied to both consonant and tone tasks; but in Experiment 2, individualized SNRs separately for consonant and tone were estimated and applied. Plosive consonants are more easily masked by background noise than lexical tones, as these consonants are characterized by aspiration noise resembling the mask, but lexical tones are featured by the fundamental frequency different from spectrum noise. In this scenario, consonant tasks should be much more difficult than lexical tone tasks and this could be another reason for the inconsistent patterns we observed in Experiment 1. For Experiment 2, however, the balancing method may still not be sufficient to overcome such a difference. Presumably, consonant tasks were still more difficult than tone tasks in Experiment 2 and thus were more susceptible to cTBS. However, as we did not directly compare lexical tone and consonant perception, this asymmetry of task difficulty may not affect our major conclusions. We added the statement to the Results of Experiment 2 (Line 190–196):

“Note that cTBS had a more extensive impact on consonant perception than on lexical tone perception. Despite our efforts to balance the individualized SNR levels for

both tasks (see Methods, Stimuli presentation), plosive consonants (featured by the length of the aspiration noise resembling the masking noise) are less resilient to noise and more challenging to identify in noise (most participants performed unsatisfactorily in the consonant-in-noise perception task; see Supplementary Fig. 2)”.

Reviewer #3:

The manuscript reports a TMS experiment in which either the putative tongue motor cortex or putative larynx motor cortex were virtually lesioned and participants performed perceptual tests related consonant or lexical pitch perception. The methodology is of exceptionally high quality and is reported explicitly and clearly. The authors conclude that virtual lesions to these motor areas impair relevant perception. However, I have several concerns about interpretation which are detailed below. Most notably, the brain coordinates at which TMS was performed correspond poorly with the known locations of the brain areas that the authors purport to have studied. This presents a considerable obstacle as it is unclear at this time whether the authors have studied the same brain regions as the literature against which they have compared their findings. This is an otherwise impressive and soundly conducted experiment that may be publishable with major revisions.

Response:

Thank you very much for your recognition of the manuscript, and we admit that the targets we selected for the tongue and laryngeal motor areas are spatially different from the frequently used locations in previous studies. Nevertheless, the targets were set to optimize the coverage of the TMS effects upon the two regions (tongue and laryngeal areas) defined by BOLD activations of our fMRI localization pretest, and the BOLD activations covered the targets used by previous studies along the vertical axis. We explain in detail in the following text.

Meanwhile, we re-emphasized our research questions and stated that in addition to the effector-specificity of the engagement of the laryngeal motor cortex, the current study also concentrated on a general question about the functional asymmetry of bilateral motor cortices in speech perceptual decision and on the temporal dynamics of the motor engagement. In this scenario, the function of the LMC is a window to speculate these broader questions. See Line 46–52: “However, three outstanding questions regarding the role of the bilateral motor cortices in speech perception remain unanswered: 1) whether the laryngeal motor cortex (LMC) is engaged in an effector-specific manner similar to the lip and tongue areas, 2) how the bilateral motor cortices cooperate during speech perception under varying difficulty, and 3) what specific stages of the perceptual decision-making process the bilateral motor cortices modulate”. The Results and the Discussion sections were reorganized correspondingly. We hope that our revised manuscript as well as the following response have better explained our perspectives and addressed all your concerns.

Major comments

Throughout - Is it insufficiently clear whether the manuscript concerns primary motor cortex (central sulcus, BA 4) or premotor cortex (precentral gyrus, BA 6). It is a habit of the TMS/perception literature to remain ambiguous on this point, but the distinction is an important one that I would urge the authors to be explicit about throughout the manuscript.

Response:

We used the BioImage Suite mapping tool (<https://bioimagesuiteweb.github.io/webapp/mni2tal.html>) to localize the "larynx" (MNI: [± 40 , -5, 50]) and "tongue" (MNI: [± 59 , -3, 36]) targets in a Brodmann area map and found that both are located in the premotor cortex (BA6) but at the posterior border nearby the primary motor cortex.

The "larynx" (or [a]) area:

The "tongue" (or [t]) area:

For humans and non-human primates, a premotor cortex is suggested to be the end of sensorimotor transforming pathways that connect the dorsal and ventral streams of vocal processing (Jarvis, 2019). It is connected to the vocal areas in the primary motor cortex. Nuttall et al. (2018) then suggested that, in speech perception, the premotor cortex functions as an intermediate region that transforms auditory input to articulatory templates when the tasks are difficult. Indeed, in addition to revealing a motor somatotopy in phoneme perception by TMS on the primary motor cortex (e.g., D'Ausilio et al., 2009), TMS on the premotor cortex also impaired phonological processing (Krieger-Redwood et al., 2013; Nuttall et al., 2018) but not semantic comprehension (Krieger-Redwood et al., 2013). In the current study, we did not, however, consider strictly the differences between the premotor cortex and the primary motor cortex. Because we wanted to ensure the spatial dissociation between TMS effects on the “Say AH” activation area and the “Say D” activation area, we applied a relatively data-driven approach by constraining our selections within the precentral gyrus that includes both BA4 and BA6. It is very possible that stimulating the selected targets affected both the premotor cortex and the primary motor cortex, and therefore

our interpretations of the functional distinctions between the two regions are very limited. However, this limitation would not affect our major conclusions in the updated version of the manuscript. See Line 127–132 in the Results: “Both targets were located in the premotor cortex (Brodmann area 6), which is suggested to be a transfer node in the sensorimotor transforming pathways²⁴ and may subserve auditory-motor mapping in adverse listening conditions²⁵. This localization approach optimized the effectiveness of TMS on our chosen tasks. The Euclidean distance between the two targets enabled the spatial dissociation of TMS effects on the dLMC and TMC²⁶”.

References:

- Jarvis, E. D. (2019). Evolution of vocal learning and spoken language. *Science*, 366(6461), 50–54. <https://doi.org/10.1126/science.aax0287>
- Nuttall, H. E., Kennedy-Higgins, D., Devlin, J. T., & Adank, P. (2018). Modulation of intra- and inter-hemispheric connectivity between primary and premotor cortex during speech perception. *Brain and Language*, 187, 74–82. <https://doi.org/10.1016/j.bandl.2017.12.002>
- D’Ausilio, A., Pulvermüller, F., Salmas, P., Bufalari, I., Begliomini, C., & Fadiga, L. (2009). The motor somatotopy of speech perception. *Current Biology*, 19, 381–385. <https://doi.org/10.1016/j.cub.2009.01.017>
- Krieger-Redwood, K., Gaskell, M. G., Lindsay, S., & Jefferies, E. (2013). The selective role of premotor cortex in speech perception: a contribution to phoneme judgements but not speech comprehension. *Journal of Cognitive Neuroscience*, 25, 2179–2188. <https://doi.org/10.1162/jocn>

L 662 -TMS motor thresholds were estimated using standard and well accepted protocols. However, it seems that thresholds were calculated on a day separate from experimental testing. These thresholds are known to vary considerably between sessions as they are sensitive to small differences in localisation (how closely hand M1 was targeted), coil orientation (typically peaking at 45 degrees from the midline), as well as participant biological factors (sleep, caffeine, etc). Please elaborate on to what extent these thresholds should be taken as valid indicators of sensitivity to stimulation across test days and to what extent this should influence the interpretation of the results.

Response:

We did take into consideration the temporal variability of the motor threshold (MT), and we chose to apply one MT testing on a day before the formal experiments for the following reasons.

1. Since what we cared about was not how TMS changes the cortical excitability of the motor cortex but whether modulating the motor cortex would affect speech perception, we individualized stimulation intensity merely for minimizing subjects' sufferings while keeping our stimulation valid.

2. Technical factors that affected the estimation of MT (but not the MT itself)

include but are not limited to the coil position and coil angle (direction of the magnetic field), but these factors are irrelevant to the testing time. It is safe to assume that the position and angle of coil which were real-time tracked by the Brainsight system should remain constant and keep the target being focused.

3. Indeed, participants' biological factors (sleep or mental states) may affect the MT. However, recent studies have provided evidence that when technical factors are fixed, participants' (resting) MT, as well as the cortical excitability, remain constant (i.e., not vary significantly) at different time points within a day (Ter Braack et al. 2019) and across different days (Therrien-Blanchet et al., 2022).

4. If we applied the MT estimation soon before the formal experiment, TMS effects during the MT assessment would remain and interfere with the stimulation protocol within the formal testing. This is because testing RMT/AMT itself requires repeatedly applying single-pulse TMS and this stimulation could modulate the excitability of the motor cortex. To avoid this interference, we placed the MT assessment on a separate day preceding the formal experiments.

5. If we adjusted the intensity of stimulation according to MT for each session of the experiment, we would add a new confounding factor (i.e., stimulation intensity) that may blur the observations.

6. We applied a Latin-square design to control the order of sessions (i.e., stimulation protocols and hemisphere) across subjects to minimize potential effects from varying cortical excitability. If the cortical excitability varies randomly, we should expect that such effects would be averaged by repeatedly testing different subjects. If, however, the changes in the excitability had a linear trend (due to weather or other population-level activities), we should expect that the effects would not influence our observations since we counterbalanced the order of different sessions.

Response:

Ter Braack, E. M., de Goede, A. A., & van Putten, M. J. A. M. (2019). Resting motor threshold, MEP and TEP variability during daytime. *Brain Topography*, 32(1), 17–27. <https://doi.org/10.1007/s10548-018-0662-7>

Therrien-Blanchet, J.-M., Ferland, M. C., Rousseau, M.-A., Badri, M., Boucher, E., Merabtime, A., Hofmann, L. H., & Théoret, H. (2022). Stability and test-retest reliability of neuronavigated TMS measures of corticospinal and intracortical excitability. *Brain Research*, 1794, 148057. <https://doi.org/10.1016/j.brainres.2022.148057>

L 611 – “48 participants performed larynx-related (say [a]) and tongue related (say [th

612] articulation tasks (dLMC: [± 40 , -5, 50], TMC: [± 59 , -3, 36]”. I commend the authors on pre-localising targets for your test population, however these coordinates are very surprising. Based on the prior literature I would have expected to find the dLMC at approximately either [-42, -12, 38] or [-50, -6, 46] for the primary motor cortex (deep in the central sulcus, BA 4) or pre-motor cortex (gyral surface, BA 6), respectively. Likewise, I might have expected tongue motor cortex to be quite a lot more ventral at about [54, -8, 28]. Given the coordinates reported in the paper, I might have expected this to be more akin to M1-lip.

Admittedly, those estimates are from samples that skew towards Caucasian and there are known differences between Caucasian as compared to Asian template brains, it seems unlikely that differences in template alone would cause such large disparities in localisation. While clearly the localisation study indicates that this coordinate is relevant for vowel production, it is not clear that dLMC is a credible interpretation.

Response:

Thank you for examining the coordinates of the targets in detail. Spatial differences indeed exist between our selected targets and the suggested targets. We will show that stimulation sites in our study also covered the suggested regions, how our choice may balance among the effectiveness of the motor somatotopy (i.e., TMC and dLMC), the optimized coverage of TMS effects, the spatial dissociation between stimulation targets, and the bilateral symmetry in the following text.

The criteria were given in the fMRI data analysis section in the Supplementary file (Line 230–236): “Criteria of target selection included: (1) bilateral homologous targets should share a similar degree of activation; (2) geometric middle points, instead of peaks of activation areas, were selected to ensure that magnetic stimulation covers the maximal volume of activations; (3) the dLMC and TMC targets should be spatially dissociated that TMS on either site would not affect the other (i.e., out of the 1.5 cm effective radius of the 70 mm figure-of-eight coil we used⁹)”.

We plot the brain activations in our functional localization pretest and the suggested targets in the following graph (panels a, b, and c) (Brain activation maps have been updated to the GitHub link https://github.com/Baishenliang/TMS_dLMC_speech/tree/main/fMRI%20pretest, as was stated in the Data Availability). Also, we plot the coordinates of the targets we used in the current study and the suggested targets (panel d) in the motor cortex. The results are shown in the figure below.

Figure. Comparisons among brain activations (orange for “say AH” – “say D”, green for “say D” – “say AH”) in the functional localization pretest, and coordinates of the targets used by the current study and the suggested targets. (a) Brain activations and the suggested dLMC in BA6; (b) Brain activations and the suggested dLMC in BA4; (c) Brain activations and the suggested TMC; (d) The spatial relationship among the used and the suggested targets.

According to a simulation study in physics, the effective coverage ($> 1/2$ maximum magnetic field strength) of the 70 mm figure-of-eight coil is 15 cm^2 , with a radius of 1.5 cm; the effective depth of the coil is ($> 1/2$ maximum magnetic field strength) around 1.5 cm (Deng et al. 2013).

Given the spatial relationship between brain activations and the suggested/used targets, as well as the effective coverage and the depth of TMS, we then illustrate how our stimulation also covered the suggested targets and simultaneously kept the spatial dissociation of stimulation.

1. Along the z (superior-inferior) axis, both the suggested BA6 ([-50, -6, 46], panel a in the Figure) and BA4 dLMC ([-42, -12, 38], panel b in the Figure) are proximate to the activations for the “say AH” tasks.

2. The suggested BA6 dLMC is close to our dLMC target (panel d in the Figure). It is slightly inferior (1 cm) to our target but is still within the effective radius (1.5 cm) of the 70 mm figure-of-eight coil when we applied TMS on our dLMC target (according to Deng et al. 2013).

3. The suggested BA4 dLMC is deeply located (panel b in the Figure) and thus may unlikely be modulated by our coil (the effective depth of stimulation is less than 1.5 cm according to Deng et al. 2013).

4. The suggested TMC is much inferior to our TMC target (panel d in the Figure) but is still close to the activations in our “say D” tasks (panel c in the Figure). We used a more dorsal TMC target because we wanted to ensure bilateral activations at the selected transverse plane (as we found fewer ventral activations for the “say [t]” task in the right precentral gyrus, panel c in the Figure).

5. Indeed, the suggested BA4 dLMC is close to our used TMC target in the z-axis (panel d in the Figure). However, the BA4 dLMC is deeply located, and the distance between them (2 cm) is beyond the effective radius of our coil (according to Deng et al. 2013). Thus, it is safe to apply TMS upon our current TMC target to modulate the activated areas in the “say D” tasks while not affecting the BA4 dLMC greatly.

On the other hand, according to Brown et al. (2008), the MNI peak coordinates of the left dorsal activation in phonation tasks (dorsal larynx/phonation area, LPA) are [-51, 0, 44] in BA6. This is similar to the suggested BA6 dLMC target and is also within the effective radius of our TMS procedure. We added **Supplementary Fig. 4** for illustration.

Supplementary Fig. 4. Group-level activation in the functional localization experiment and the dLMC target location.

(a–c) Group-level activation maps. The axial views show the positions of dLMC (**a and b**, “say AH” – “say D”, orange) and TMC (**b**, “say D” – “say AH”, green) activation areas. (**c**) Dorsal-ventral functional dissociation between larynx (orange) and tongue (green) motor areas is shown from the coronal view of activation.

(d–i) Comparison between the location of the used dLMC target in the current study and the activation peak for the dorsal larynx/phonation area (LPA) in Brown et al. (2008)¹⁰. (**d**) The dLMC target and the dorsal LPA peak on the surface map (red dots) with the activation volumes for dLMC (“say AH” – “say D”, colored areas). Note that, both nodes are within the dorsal-ventral range of the activations. (**e and f**) The spatial relationship between the dorsal LPA (red dot) and the “say AH” – “say D” activation areas (orange areas) from the coronal (**e**) and sagittal (**f**) views. (**g–i**) The spatial relationship between the dLMC target (red dot) and the “say AH” – “say D” activation areas (orange areas) from the axial (**g**), coronal (**h**), and sagittal (**i**) views.

Activation areas were masked by AAL ROIs of bilateral precentral gyrus. L, left hemisphere; R, right hemisphere. x, y, and z represent MNI coordinates at the left-right, anterior-posterior, and inferior-superior extents, respectively. Visualization was performed by Mango¹³.

In conclusion, we admit that our target selection strategy differed from most TMS

studies in the relevant topics. We added this limitation in the Discussion (Line 427–435): “Limitations exist in the current study regarding the location and function of the dLMC. Firstly, the dLMC stimulation targets we defined ($z = 50$) were superior to the activation peak of the dorsal larynx/phonation area (LPA) found by Brown et al. (2018) ($z = 44$)¹⁴. However, while we selected the dLMC targets based on brain activations that recruited laryngeal functions, the previously found dorsal LPA is within the ventral-dorsal range of activations for vowel production in the current study and the effective radius (1.5 cm) of TMS²⁶ (See Supplementary Fig. 4). Still, future studies are needed to test the effects of tonal language experience on the location of the dLMC, as previous localization work has been biased towards non-tonal language users”.

Response:

Brown, S., Ngan, E., & Liotti, M. (2008). A larynx area in the human motor cortex. *Cerebral Cortex*, 18, 837–845. <https://doi.org/10.1093/cercor/bhm131>

Deng, Z. De, Lisanby, S. H., & Peterchev, A. V. (2013). Electric field depth-focality tradeoff in transcranial magnetic stimulation: simulation comparison of 50 coil designs. *Brain Stimulation*, 6(1), 1–13. <https://doi.org/10.1016/j.brs.2012.02.005>

The interpretation in the discussion section is difficult to evaluate without greater certainty around localisation.

Response:

We have revised the Introduction and the Discussion sections to better organize the key points of the current study, and hope that the new Discussion section has addressed all issues that you are concerned about.

L424 – This paragraph makes several claims that are vague, over-reaching and/or not supportable by the references provided. Please revise.

Response:

We revised the Discussion section and interpreted the functional asymmetry of the motor engagement in lexical tone and voicing perception in response to our hypotheses. Particularly, for the discovery of the recruitment of bilateral dLMC in VOT identification, we stated that it provides evidence to support the “motor hypothesis” (i.e., the motor cortex is engaged in speech perception as in production) and weakly supports the “lexical hypothesis” (i.e., lexical experience drives the hemispheric asymmetry of the motor engagement). See below for the revised contents in the

Discussion:

Line 333–348: “Previous TMS studies have reported that the dLMC is causally involved in the discrimination of non-lexical vocal pitch^{10,31} and singing voice³². A recent study demonstrated that lexical tone processing is suppressed after stimulating a lip region close to the dLMC in the left motor cortex³³. The current study takes a step further and establishes a causal link between the human dLMC (defined by BOLD activations in articulation tasks) and perceptual decision of lexical tone. In addition, we provide previously unreported evidence that bilateral dLMC were causally recruited in detecting subtle VOT differences for plosive consonants. This parallels the dLMC’s motor function to accurately control the moment of voicing onset. Notably, the consonant results are inconsistent with Sammler et al. (2015)¹⁰ as they found that stimulating the dLMC in neither side affected consonant perception. However, our tasks were more challenging due to extra noise masking, which might urge the dLMC engagement. Overall, previous studies have reported that the right motor cortex entrains speech stream³⁴ and is involved in auditory-speech coupling³⁵ resembling its left counterpart, and we provide new evidence for its engagement in the perception of lower-level phonemic cues”.

Lines 375–377: “In contrast, our results only provide weak support for the lexical hypothesis, which suggests that the functional asymmetry of the motor cortex is due to differences in linguistic functions between lexical and non-lexical speech cues”.

Discussion – The focus here on the dorsal stream (usually conceived of as STG->IFG via the arcuate fasciculus) and the ventral stream (usually conceived of as the ITG/MTG -> IFG via the uncinata fasciculus) does not seem well motivated. The manipulations in the present study are at (pre?)-motor cortex and well downstream of this distinction. I do not see how this is a particular issue on which the present study provides new information.

Response:

Indeed, the current study concentrated on the effector-specific involvement of bilateral dLMC in speech perceptual decision, whereas dual-stream models emphasize parallel processing streams for the sound-to-articulation projection (dorsal stream) and the sound-to-meaning transformation (ventral stream). Nevertheless, as mentioned by Hickok & Poeppel (2007), the premotor cortex targeted on by the current study belongs to the dorsal stream. Moreover, they proposed that the dorsal stream is strongly left-lateralized, but the current study gives new evidence for the recruitment of the right

motor cortex, as well as the temporal dynamics of motor engagement.

At the beginning of the Introduction in the revised manuscript, we listed the dual-stream models along with the motor theory and the motor somatotopy in speech perception as background theories, and pointed out the current gaps of knowledge (Line 38–52):

“Speech perception has long been hypothesized to recruit motoric simulation by the speech motor system, as posited by the motor theory of speech perception¹. Recent neuroanatomical models of speech processing propose that the left motor cortex maps phonological analyses onto motor representations, which may compensate for degraded auditory processing in challenging listening conditions^{2–4}. Transcranial magnetic stimulation (TMS) studies have identified a causal engagement of the left motor cortex in speech perception in an effector-specific manner, such as the lip motor subregion for bilabial consonants^{5,6}, and the tongue motor area for dental consonants^{7,8} and vowels⁹, while the right motor cortex has been linked to non-lexical prosodic cues¹⁰. However, three outstanding questions regarding the role of the bilateral motor cortices in speech perception remain unanswered: 1) whether the laryngeal motor cortex (LMC) is engaged in an effector-specific manner similar to the lip and tongue areas, 2) how the bilateral motor cortices cooperate during speech perception under varying difficulty, and 3) what specific stages of the perceptual decision-making process the bilateral motor cortices modulate.”.

We then proposed four hypothetical mechanisms that may drive the functional distributions of bilateral motor cortices in speech perception (see Fig. 1 and the Introduction for details).

We responded to the dual-stream models by listing our main findings and indicating how these findings may provide new knowledge within the framework of the current models in the Conclusion (Lines 456–467).

“In conclusion, as illustrated in Fig. 5, we propose a model of bilateral dLMC engagement in the temporal dynamics of perceptual decision for lexical speech cues. Firstly, the recruitment of bilateral dLMC for the perceptual decision of Mandarin lexical tone and voicing of consonant confirms the motor hypothesis regarding bilateral effector-specificity. Secondly, the left-dominance of the dLMC and the compensatory engagement of its right counterpart under increased difficulty support the redundancy hypothesis. Thirdly, bilateral dLMC are involved in multiple stages of the perceptual decision-making process, which is mediated by the hemisphere and cognitive demands.

This study provides strong support for the sensorimotor integration account of speech perception^{2,3}. Moreover, our findings broaden the scope of research on this topic by providing insight into the functional spatial distributions of bilateral motor cortices and their involvement in the temporal dynamics of speech perceptual decision-making.”

Minor comments

Figure 1a. The “ “ around “mirrors” leads one to suspect that this is an allusion to mirror neurons. As this concept has been habitually overstated and under-evidenced I would suggest omitting it here unless the authors wish to make an explicit case that they are studying mirror neurons. The figure itself is excellent.

L466 knowledge

L 818 Fourth.

Response:

Thank you for pointing out these writing problems. We have removed the term “mirrors” in Fig. 1, as well as in the figure caption. We also corrected the mentioned spelling mistakes and checked through the manuscript for typos.

REVIEWERS' COMMENTS

Reviewer #1 (Remarks to the Author):

The authors have addressed all my concerns.

Reviewer #2 (Remarks to the Author):

The authors made important changes to their manuscript to better focus the research question and to better frame their findings in the relevant literature.

The revision has also improved significantly in terms of clarity, with the addition of new figures.

However, my concern about the role of Exp1 and the imbalance between consonant and lexical tone tasks remains in part.

In this regard, the authors have toned down the use of Exp1 in discussions and provided explanations for the task imbalance (which is present in both Exp).

In conclusions, for such a high-profile journal, I feel a bit perplexed to suggest acceptance. Indeed, the pattern of results is not crystal clear, as I would imagine is essential for a journal of this type.

Reviewer #3 (Remarks to the Author):

The authors have addressed all of my comments and I therefore recommend the article for publication. Below I list a few minor comments which may further improve the manuscript.

L289 ...(or more generally, right motor cortex) is redundant *for this task* and offers compensation in adverse listening contexts.

L427+ The reference to Brown 2008's LPA is well taken and this reviewer understands that this is simply an alternate terminology for the dorsak LMC which the authors refer to throughout. However, this correspondence may be lost on readers unfamiliar with that literature. They may also stumble on the implied contrast to a ventral LMC that is not discussed in the manuscript (quite justly as it was not observed in the imaging experiments). Clearly an extensive discussion about nomenclature would be out of place, but you might consider referring to more detailed discussions found elsewhere, e.g. in any of the following:

Belyk, M., & Brown, S. (2017). The origins of the vocal brain in humans. *Neuroscience & Biobehavioral Reviews*, 77, 177–193. <https://doi.org/10.1016/j.neubiorev.2017.03.014>

Belyk, M., Eichert, N., & McGettigan, C. (2021). A dual larynx motor networks hypothesis. *Philosophical Transactions of the Royal Society B: Biological Sciences*, 376, 20200392. <https://doi.org/10.1098/rstb.2020.0392>

Eichert, N., Papp, D., Mars, R. B., & Watkins, K. E. (2020). Mapping human laryngeal motor cortex during vocalization. *Cerebral Cortex*, 30(12), 6254–6269. <https://doi.org/10.1101/2020.02.20.958314>

L433 resistant instead of reluctant?

We thank the editor and three reviewers for giving constructive advices. To account for the critiques and suggestions, we have revised the manuscript. The following text shows our point-by-point responses to your concerns and questions. With your help, we believe that the manuscript has been improved and can better demonstrate our main discoveries, and we hope that all the concerns have been addressed satisfactorily. In the revised manuscript, all changes are marked in red. Thank you for the opportunity to resubmit our work.

Reviewer #1:

The authors have addressed all my concerns.

Response:

Thank you.

Reviewer #2:

The authors made important changes to their manuscript to better focus the research question and to better frame their findings in the relevant literature.

The revision has also improved significantly in terms of clarity, with the addition of new figures.

However, my concern about the role of Exp1 and the imbalance between consonant and lexical tone tasks remains in part.

In this regard, the authors have toned down the use of Exp1 in discussions and provided explanations for the task imbalance (which is present in both Exp).

In conclusions, for such a high-profile journal, I feel a bit perplexed to suggest acceptance. Indeed, the pattern of results is not crystal clear, as I would imagine is essential for a journal of this type.

Response:

Thank you very much for your recognition of the improvement.

Although Exp1 is exploratory, it validated our stimulation protocol and showed potential effects that were expected.

Meanwhile, we made our main conclusions based on Exp2, which had imbalance results between lexical tone and consonant tasks but such imbalance did not influence our main conclusions.

Reviewer #3:

The authors have addressed all of my comments and I therefore recommend the article for publication. Below I list a few minor comments which may further improve the manuscript.

L289 ...(or more generally, right motor cortex) is redundant *for this task* and offers compensation in adverse listening contexts.

L427+ The reference to Brown 2008's LPA is well taken and this reviewer understands that this is simply an alternate terminology for the dorsal LMC which the authors refer to throughout. However, this correspondence may be lost on readers unfamiliar with that literature. They may also stumble on the implied contrast to a ventral LMC that is not discussed in the manuscript (quite justly as it was not observed in the imaging experiments). Clearly an extensive discussion about nomenclature would be out of place, but you might consider referring to more detailed discussions found elsewhere, e.g. in any of the following:

Belyk, M., & Brown, S. (2017). The origins of the vocal brain in humans. *Neuroscience & Biobehavioral Reviews*, 77, 177–193. <https://doi.org/10.1016/j.neubiorev.2017.03.014>

Belyk, M., Eichert, N., & McGettigan, C. (2021). A dual larynx motor networks hypothesis. *Philosophical Transactions of the Royal Society B: Biological Sciences*, 376, 20200392. <https://doi.org/10.1098/rstb.2020.0392>

Eichert, N., Papp, D., Mars, R. B., & Watkins, K. E. (2020). Mapping human laryngeal motor cortex during vocalization. *Cerebral Cortex*, 30(12), 6254–6269. <https://doi.org/10.1101/2020.02.20.958314>

L433 resistant instead of reluctant?

Response:

Thank you for your recognition and your comments.

For the LPA and dLMC (and the vLMC), we have shown the correspondence between the term “LPA” and “dLMC”. Line 429: “dorsal larynx/phonation area, (LPA, corresponding to the dLMC)”, and added the following content along with the recommended papers (lines 433–440):

“In addition, research has found two subregions in the human motor cortex to control laryngeal movements: the ventral and dorsal LMC¹⁴, belonging to separate laryngeal movement control networks⁴⁹. The vLMC is a homologous area of the non-human primate LMC, while the dorsal region may have evolved as a human-specific

area⁵⁰. Although the functions of these two LMCs are not fully understood, the dLMC may be closely associated with complex human language and singing abilities¹⁰. Nevertheless, future study should still compare functions of the two LMCs in speech perception.”

The mentioned minor problems have been solved:

Lines 288–289: “...(or more generally, right motor cortex) is redundant **for this task and...**”

Line 451: “...fully recruited and **resistant** to further enhancement...”